# Circadian-Mediated Regulation of Growth, Chloroplast Proteome, Targeted Metabolomics and Gene Regulatory Network in *Spinacia oleracea* Under Drought Stress

Ajila Venkat [1,2] and Sowbiya Muneer [1,*]

1   Horticulture and Molecular Physiology Laboratory, Department of Horticulture and Food Science, School of Agricultural Innovations and Advanced Learning, Vellore Institute of Technology, Vellore 632014, India; ajilavenkat@gmail.com
2   School of Biosciences and Technology, Vellore Institute of Technology, Vellore 632014, India
*   Correspondence: sowbiya.muneer@vit.ac.in or sobiyakhan126@gmail.com

**Abstract:** The paramount objectives of this study were to analyze the beneficial role of the circadian clock in alleviating drought stress in an essential green leafy horticultural crop, spinach (*Spinacia oleracea*), and to attain knowledge on drought-stress adaptation for crop productivity. From dawn to dusk, a circadian core oscillator-based defense mechanism was noticed in relation to the strength of the chloroplast proteome and transcriptome, and the defense hormone fused it along with the molecular physiology using genotypes "Malav Jyoti" and "Delhi Green". A photo-periodic rhythmicity containing a 4 h time interval (morning–evening loop) for 12 h in spinach was exhibited under drought-stressed (day-5) and drought re-irrigated (day-10) conditions. The circadian oscillator controlled 70% of the major part of growth and physiological measures such as the biomass, plant height, leaf-relative water content, and the shoot–root ratio under drought stress. Contrarily, drought stress resulted in the upregulation of antioxidative activities and stress markers, whereas it was diversified and maintained in the case of the re-irrigated state at certain rhythmic time intervals of the circadian clock. The physiological parameters we examined, such as net photosynthesis, transpiration, stomatal conductance, and antioxidative enzymes, exhibited the role of the circadian clock in drought stress by showing 80–90% improvements found in plants when they were re-watered after drought stress based on their circadian oscillations. Based on the physiological results, 10 a.m. and 2 p.m. were disclosed to be the rhythmic times for controlling drought stress. Moreover, an extensive study on a gene expression analysis of circadian clock-based genes (*CCA1*, *LHY*, *TOC1*, *PRR3*, *PRR5*, *PRR7*, *PRR9*, and *RVE8*) and drought-responsive genes (*DREB1*, *DREB2*, and *PIP1*) depicted the necessity of a circadian oscillator in alleviating drought stress. Hence, the findings of our study allowed for an intense understanding of photo-periodic rhythms in terms of the morning–evening loop, which is in line with the survival rate of spinach plants and occurs by altering cellular ROS-scavenging mechanisms, chloroplast protein profiles, gene regulation, and metabolite concentrations.

**Keywords:** drought stress; plant circadian rhythm; ROS scavenging mechanism; thylakoidal proteome; targeted metabolomics; gene-regulatory network



## 1. Introduction

Drought stress has become a major destructive factor among all abiotic stresses, which has led to an unfavorable decrease of 45% in the development of agricultural crop production throughout the world [1]. Additionally, drought stress worsens the quality of

grain and crop yields as a consequence of alterations in plant physiological functions [2]; furthermore, drought stress leads to biochemical modifications and physical damage at the overall organism and cellular levels [3]. Drought also has the capability to block photosynthesis, respiration, and stomatal movement, which are further constraints in plant growth and physiological metabolism [4]. Drought resistance is a wide term that includes certain adaptive mechanisms or features, such as drought escape, drought tolerance, and drought avoidance, that plants use to manage the effects of drought stress [5,6]. An increase in the abscisic acid concentration is often noticed under drought and other abiotic stresses [7]. When a crop is sensitive to drought stress, it leads to the following effects: declined levels of photosynthesis [8] and other processes, a smaller leaf area index [9], stomatal closure, a reduction in $CO_2$ intake, an increase in photo-respiration, limited carboxylation, decreased water potential, a reduction in Rubisco activity, an increase in reactive oxygen species (ROS) [10], oxidative damage to chloroplasts and plasma membranes, the downregulation of non-cyclic transport, obstruction in ATP synthesis, a reduction in nutrient uptake, susceptibility towards disease, and decreased yield quality and production [11]. There are different types of ROS that are produced in plants, namely $H_2O_2$ (hydrogen peroxide), $O_2^{-1}$ (singlet oxygen), CO (carbon mono-oxide), $NO_x$ (nitroxide), $SO_2$ (sulfur dioxide) [12], $O_2^{\bullet-}$ (superoxide anions), and OH (hydroxyl radicals) [10,13].

The production of ROS in plants acts as an alarm signal that could further kickstart the defense mechanism or acclimatization process [14]. Sodium dismutase (SOD), ascorbate peroxidase (APX), catalase (CAT), citrullines, polyamines, and several enzymes act as antioxidants, which minimizes the effects of drought stress [15]. Ascorbic acid (AsA) and glutathione are the low-molecular-weight antioxidants that are present in nearly all organelles, and they are associated with ROS quenching, initially by lending reducing equivalents to ROS-scavenging enzymes [13,16]. Thus, this process has the capability to maintain the decreased levels of ROS concentrations even if ROS are produced at a high rate as these enzymes can directly react with ROS [17,18]. Plant growth hormones such as auxins, cytokinins, gibberellins, abscisic acid, and salicylic acid alter plant responses to drought stress [15]. Defense hormones such as jasmonic acid (known as a first defense hormone) and its derivatives, such as methyl ester (MeJA) and isoleucine conjugate (JA-Ile), are known together as jasmonates. They play a major regulatory role in different aspects of plant growth and development [19]. JA, JA-Ile, cis-OPDA, ABA, and SA have the ability to improve stress tolerance in plants; along with this, SA helps plants in abiotic stress mitigation [20]. ABA is an essential mediator of drought stress, and it has the capability to regulate plant water balance and provide osmotic stress tolerance [21]. OPDA (precursor of JAs) has a vital role in drought tolerance through various signaling pathways, and it remains different from JA-Ile [22].

The circadian clock in plants is often known to be a central oscillator or internal biological timekeeper that helps plants receive environmental cues such as light, humidity, nutrients [23], and temperature (the input pathway has the capability to reset the clock) [24,25], and it supports roughly 24 h of oscillation (16 h light and 8 h dark) in the expression of genes, which manages the changes with output genes [26]. Moreover, the circadian clock helps in managing photo-periodic rhythmicity for the enhanced growth, development, and fitness of plants [27]. Over the last few decades, it has been revealed that the circadian oscillator in plants has the ability to provide them stability towards the adaptation or alleviation of stress [28]. The clock consisting of a self-sustained mechanism and its various transcriptional and translational feedback loops (TTFLs) are well maintained in plant species [27,29]. This biological clock has a control on a set of physiological and biochemical pathways likely seed germination, photosynthesis, hypocotyl elongation, flowering, stomatal movement, respiration, ion uptake, sugar metabolism,

nutrient metabolism, translocation, phytohormones, and senescence [14,30]. The clock genes regulate each other's expression, having certain positive and negative feedback loops; also, these genes are expressed especially at a particular time period of the day [24]. The photo-periodic rhythmicity of the biological clock starts with an input pathway where the external environmental cues, such as light (daily dawn and dusk = *zeitgeber*) [31] and temperature, pass the light signals [32] to photo-receptors, namely phytochromes (PHYs) and cryptochromes (CRYs) [33,34]. Then, these light stimuli are transported (light-signal transduction) and integrated into the clock [35].

Spinach (*Spinacia oleracea*) is a green leafy plant (economically crucial crop) and has its native origin in the central and southwestern regions of Asia [36,37]. It belongs to the order Caryophyllales, the family Amaranthaceae, and subfamily Chenopodioideae [38–40]. The leading producers of *Spinacia oleracea* L. in India are Telangana, Kerala, Tamil Nadu, Karnataka, Maharashtra, West Bengal, Andhra Pradesh, and Gujarat [41]. The optimum range of temperature for growing spinach is between 15–30 °C, along with a rainfall of about 80 to 120 cm; usually, they grow up to the height of 30 cm (along with a stem of 50–90 cm high [42]) and are nearly 15 cm wide during harvesting stage [41]. Spinach is a well-known long-day plant; the arrangement of *S. oleracea* appears to be alternate, and the true leaves are formed during days 4–11 after seedlings [42]. A rapid and sudden change in climate, along with reduced availability of water, restricts spinach cultivation [43]; the production of cultivated spinach per annum is 30 million tons worldwide [44]. Spinach leaves have a rich source of vitamins of the B group, ascorbic acid, micro-nutrients, phytonutrients, and beta-carotene (or pro-vitamin A) [40,41]. The seeds of spinach genotypes are of various forms like round, pointed, or prickly, and the texture of the leaf is flat, smooth, semi-savoy, or crinkled (=savoy) [39]. Also, there are early and late genotypes that may vary promptly during stem elongation according to the leaf rosette [40]. It has a shallow root architecture, and it needs regular irrigation to maintain the soil moisture content for constant leaf growth and development [45].

The current research was orchestrated with the aim to explore the effect of drought stress on horticultural crop spinach (*Spinacia oleracea* L.) genotypes (Delhi Green (DG) and Malav Jyoti (MJ)) on the basis of 12 h of photo-periodic rhythmicity at the time intervals of 4 h; this is exactly by 6 and 10 a.m. and 2 and 6 p.m. on day 5 (stressed period) and day 10 (drought re-irrigated period). The focus of this research was to furnish an overview of the investigation of drought-resistant and drought-sensitive spinach genotypes with specific measurements such as morpho-physiological and photosynthetic variables, oxidative parameters, cytotoxic analysis, phytochemicals (phenol), protein, and thylakoidal proteome analysis. Moreover, there are no preceding studies on the impact of drought stress on spinach cultivars at the cellular level or ROS-scavenging mechanism, nor are they accompanied by data of molecular-level analyses of phytohormones concentrations along with circadian and drought gene-regulatory networks (having *CCA1*, *LHY*, *TOC1*, *RVE8*, *PRR3*, *PRR5*, *PRR7*, *PRR9*; and *DREB1*, *DREB2*, *PIP1*) in spinach genotypes on day 5 and day 10 (10 a.m. and 2 p.m.). From our findings, we hypothesized that the circadian core oscillators play a vital role, especially in alleviating drought stress by altering the physiological parameters such as photosynthesis, antioxidative mechanisms, chloroplast proteome, and the gene-regulatory network along with targeted metabolomics, mainly during particular periods (10 a.m. and 2 p.m.) of the circadian cycle.

## 2. Materials and Methods

### 2.1. Growth Environment and Collection of Plant Samples

The current research was performed in the polyhouse of the School of Agricultural Innovations and Advanced Learning, Vellore Institute of Technology, Vellore, India. The

polyhouse had growth conditions that are as follows: a dawn and dusk period of 16 h: 8 h; temperature regime of 30 °C during dawn and 25 °C after dusk; and the relative humidity was around 65 ± 5%. Spinach (*Spinacia oleracea* L.) genotypes such as 'Delhi Green' and 'Malav Jyoti' were bought from a seed retailer—the Indian Government seed shop (Rajamanickam Agro Service). These two genotypes were selected based on their popularity among local growers and consumers' acceptance because of their large leaf areas. The spinach seeds underwent a surface sterilization process, having 5% sodium hypochlorite (NaClO) solution for about 30 min; later, it was rinsed with distilled water. The grow bags were obtained from Bio Blooms, Coimbatore city, India, with the dimensions of 40 cm × 24 cm × 24 cm (length × height × width). The grow bags were filled with 1:1 red soil with vermicompost in a respective ratio (filled up to 3 kg, though it had the capacity to accommodate 4.5 kg, and the top ends of all the grow bags used were kept folded) with a soil pH of 6.86.

### 2.2. Experimental Layout and Drought Stress Treatment

The experimental setup was carried out consisting of 24 grow bags filled with the mentioned soil composition; these 24 grow bags were then separated into 2 sets, each having 12 grow bags for Delhi Green (DG) and 12 grow bags for Malav Jyoti (MJ). Thus, 6 grow bags were maintained for each treatment (MJ-Control (MJ-C), MJ-Stress (MJ-S), DG-Control (MJ-C), and DG-Stress (DG-S)). Each grow bag was sown with 10 spinach seeds (at 1 cm of depth from the upper surface of the soil bed), and a further thinning process was carried out after germination. The spinach plants were permitted to grow until they reached the vegetative stage (approximately between 27–30 days after germination). To treat the spinach plants with drought stress, the grow bags with MJ-S and DG-S were withheld from irrigation for 6 days; thus, the day of water withholding commencement (a chemical-free method for inducing drought treatment) was considered as day 0. MJ-C and DG-C were irrigated normally (with 100 mL of water per grow bag) on a daily basis. Thus, day 5 was taken as a reference point for sample collection (leaf samples were collected), and from day 6, the grow bags of MJ-S and DG-S were re-irrigated, and by day 10, the samples were collected. To check the moisture content in the drought-stressed and controlled plants, the grow bags were weighed before and after treatments, in addition to moisture sensors were used to confirm the drought stress. The leaf samples were collected during day 5 and day 10 at the time interval of 4 h (with respect to the morning–evening circadian loop), which were at precisely 6 and 10 a.m. and 2 and 6 p.m.; these samples were then stored in the deep-freezer (−80 °C) after each time interval until further use. For conducting further experiments, analytical and molecular-level chemicals or substances were employed. For each experimental analysis, three biological replicates were taken from each treatment.

### 2.3. Morphological Observations and Growth Measurements

From the group of 6 grow bags with each treatment (MJ-C, MJ-S, DG-C, and DG-S), 4 grow bags of each set were kept for obtaining 4 replicates of samples for further experiments, and 2 grow bags were used for photographs.

During day 5 of drought stress and day 10 of re-irrigation, the plants were gently removed from the grow bags. The plants were then utilized for the measurement of plant height and root length on the same day. Photographs were captured for both the control and stress of both spinach genotypes to obtain the plant heights, which are provided with a scale alongside the spinach plants.

### 2.4. Plant Biomass and Leaf-Relative Water Content (L-RWC)

The 3 different plants from 3 different grow bags (biological triplicates) were taken gently without disturbing the root area for measuring the plant biomass. Then, the plants were washed using distilled water. The fresh and dry biomasses were weighed utilizing a digital weighing balance. For the dry biomass, the samples were held in a hot-air oven for exactly 42 h at 65 °C. The plant biomass was calculated according to Al Murad and Muneer [46]. The leaf-relative water content (LRWC) was determined using the standard procedure as proposed by Barrs and Weatherley [47]. Then, the obtained values were used in the formula mentioned by Razi et al. [48].

### 2.5. Estimation of Photosynthetic Variables and Chlorophyll Fluorescence

The net photosynthetic rate (NPR), transpiration rate (TR), stomatal conductance (SC), and chlorophyll fluorescence (Fv/Fm) of single leaf blade (in biological triplicates) measurements were taken around 10 a.m. of both days 5 and 10, with the help of a portable apparatus, the SPAD meter (Konika Minolta, Tokyo, Japan). Another piece of equipment, named the PAM 2000 chlorophyll fluorescence meter (Heinz Walz GmbH, Zarges 40860, Weilheim, Germany), was employed to measure the chlorophyll fluorescence (Fv/Fm). The maximum PSII-quantum yield with the unit of Fv/Fm was calculated using the formula $Fv/FM = (Fm - F0)/Fm$. For the PSII-quantum yield, leaves from each treatment were kept under the dark by placing a leaf clip for 30 min before the measurement was taken, preferably on the youngest fully developed leaf. After incubation in the dark, the sensor provided with the instrument emitted a strong light illumination and was used to take the absorbance of the PSII-quantum yield. The sensor emits a strong flash of light. The excess of light excites the chlorophyll in the leaf, which re-emits light in the form of chlorophyll fluorescence. The amount of fluorescence depends on the state of the photosystem II.

### 2.6. Estimation of Photosynthetic Pigment

The fresh leaf samples collected between 6 and 10 a.m. and 2 and 6 p.m. on both days 5 and 10 were used to perform the pigment content (the total chlorophyll content and carotenoid content) estimation as characterized in [49], and it was calculated according to the Arnon [50] formulae.

### 2.7. MDA and Proline Content (Stress Markers)

Malondialdehyde content (MDA) by the thio-barbituric acid reactive substances method (TBARS) was performed as described in previous studies [51,52]. The proline content in the spinach genotypes was estimated according to [53], with minor changes.

### 2.8. Determination of Total Protein and Total Phenol Contents

The quantitative analysis for total soluble protein was performed using the Bradford method [54] with minor modifications according to [55]. Bovine Serum Albumin (BSA) was used for a standard curve to estimate the unknown value of the total protein content present in the plant sample. The extract of the sample was then utilized for the estimation of the total protein content under a UV spectrophotometer (Sl. NO-A120656, UV-1280, Shimadzu, Kyoto, Japan) at an absorbance range of 595 nm. The total phenol content was carried out using the Folin–Ciocalteau method according to a previous study [52], and here, Gallic acid was used as a standard.

### 2.9. In Situ Localization of $H_2O_2$ and $O_2^{-1}$

The histo-chemical staining of hydrogen peroxide and superoxide radicles was performed at 10 a.m. and 2 p.m. The leaf samples were collected on day 5 and day 10, as stated by a previous study [56,57] with the slight modifications mentioned in [51]. Along

with this, the fresh leaves soon after collected were cut into a circle form with the help of a screw cap.

## 2.10. Estimation of Antioxidant Enzymatic Activities and Their Native PAGE Profiling

The leaf samples collected between 10 a.m. and 2 p.m. on both days 5 and 10 of drought-stress and re-irrigated conditions were used to determine the activities of the SOD, CAT, and APX enzymes and native PAGE profiling, as specified previously [51]. The SOD activity was detected using the nitro blue tetrazolium (NBT) inhibition method, as explained in [58], with minor modifications. The activity of the CAT enzyme in plant tissues was carried out according to [59], with minor changes. APX activity in the plant leaf samples was estimated by the method [60], with slight modifications.

Isozymes of SOD, APX, and CAT were estimated as per the protocol from [60], including slight changes. Around 30 μg of protein were used for each profiling; for the SOD, samples were prepared on the basis of [61], and the staining was performed as per [62]. CAT isozyme sample preparation was done according to [63], and the staining method was followed by [64]. APX isozymes' samples were prepared with the help of [65]; the staining was conducted using the procedure of [66].

## 2.11. Stomatal Index and Stomatal Appearance

The freshly collected spinach leaf samples (day 5 and day 10) were folded carefully, and forceps were used to detach a peeled segment from the bottom surface of the leaf. Then, the peel was left in a watch glass filled with water for a few minutes, and the sample was further stained with a few drops of safranin for around 3 min; later, a drop of glycerin was applied using a needle. The structure of the stomata was examined under high-power magnification using a compound microscope. The same samples were utilized for the stomatal index determination, as indicated earlier [49,56].

## 2.12. Thylakoidal Protein Analysis (BN-PAGE)

The native proteins present in thylakoids were analyzed at 10 a.m. and 2 p.m. The leaf samples collected during both days 5 and 10 were conducted by performing 1D BN-PAGE, according to prior research [52,67].

## 2.13. Isolation of RNA, cDNA Synthesis, and Analysis of RT-PCR

RNA was separated from the spinach leaf samples collected at 10 a.m. and 2 p.m. on both day 5 and day 10, using an RNA isolation kit, and the manufacturer's (Hi-Media, Mumbai, Maharashtra, India) instructions were followed. The quality of RNA was checked using a nanodrop at 280 nm, and a ratio of 1.8 to 2 was considered a pure form and considered for cDNA synthesis. The cDNA for all samples was synthesized using the Takara kit, having their composition of the same. Real-time PCR (RT-PCR) was performed with Bio-Rad (Hercules, CA, USA) master mix and its instructions, having the protocol of 3 min at 95 °C, then 40 cycles of 30 s of 95 °C, 30 s of 55–61 °C, followed by 5 min at 72 °C. Actin was utilized as an internal control for the normalization of all quantifications [68,69]. Three separate leaf samples from the independently grown plants were taken for RNA preparations, and two duplicates (for each sample) were used for the RT-PCR experiments (qPCR). The obtained results were analyzed with qBase plus 28 software 13. The gene-specific primers utilized in our investigation are enrolled in Tables 1 and 2.

**Table 1.** List of primer sequences utilized for the gene expression analyses through RT-PCR analysis. It consists of a set of gene names and their sequences, which were calculated using Actin-1 (internal control).

| Gene Name | Forward Sequence (5′---------3′) | Reverse Sequence (5′---------3′) | Amplicon Size (bp) |
|---|---|---|---|
| *Actin*-1 | CGTTTGGATATTTTGCCTGCC | GTAGTCTGTCAGGTCACGCC | 200 |
| *CCA1* | TTCAGCCACTAGTATGTTGA | GATAGAGAACTTGTTATTGA | 800 |
| *LHY* | GCAATCCTCGGAACCAACCA | GACTGTTTCACGGTGGACTT | 920 |
| *RVE8* | ACATCCCAACTTCTTCCCCG | CTTGAGCTTCCCATGCCAGA | 958 |
| *PRR5* | ATGACTAGTAGCGAGGAAGT | TGTTACGTCGTCCAGTTCTT | 400 |
| *PRR7* | TGGCCCGAGACAAATCTTTC | AAGATCCTGACTATTATTAT | 800 |
| *DREB1* | ATGACCTCATTTTCTACCTT | TTAATAACTCCAAAGGGACA | 645 |
| *DREB2* | GGAAGAAGTTTCGGGAGACG | CCAACAAGGTCGGCATCCCG | 429 |
| *PIP1* | ATGGAAGGGAAAGATGAAGA | TTACGACTTGGACTTGAATG | 861 |

**Table 2.** List of primer sequences utilized for the gene expression analyses through RT-PCR analysis. It consists of a set of gene names and their sequences, which were calculated using Actin-2 (internal control).

| Gene Name | Forward Sequence (5′---------3′) | Reverse Sequence (5′---------3′) | Amplicon Size |
|---|---|---|---|
| *Actin*-2 | ATCCTCCGTCTTGACCTTG | TGTCCGTCAGGCAACTCAT | 200 |
| *TOC1* | AACAAAGTCTTCTTCTTTCT | ACACTCATCTCGACTCTTGT | 600 |
| *PRR3* | ACCACCTCCCAAAAAATCC | TGGATAAACAAAAATTTAAT | 700 |
| *PRR9* | AACAAAGTCTTCTTCTTTCT | ACACTCATCTCGACTCTTGT | 500 |

*2.14. Targeted Metabolomics Analysis*

The concentration levels of 5 different defense hormones (phytohormones) were analyzed at 10 a.m. and 2 p.m. on both days 5 and 10, with the help of UHPLC-MS/MS (QTRAP 6500). For the analysis of each hormone, around 30 mg of plant leaf samples were used. The samples stored at $-20\,^\circ$C were lyophilized or freeze-dried using Lyophilizer (Model No. 4 KG/$-80\,^\circ$C, LARK, Chennai, Tamil Nadu, India), and they were dissolved in the solvent using Methanol. In order to account for variability across different metrics, all of the samples were normalized either post-acquisition (using techniques like "constant sum normalization" or specialized algorithms like NOMIS (normalization by optimal multiple internal standards)) or pre-acquisition (using a measured quantity representing the total sample amount to adjust sample volumes before analysis) using internal standards added to each sample during extraction.

The internal standards (IS) used were D6-JA for JA (jasmonic acid), [$^2$H4] salicylic acid (d4-SA) for SA (salicylic acid), [$^2$H6] (+)-cis, trans-abscisic acid (D6-ABA) for ABA (abscisic acid), D6-JA-Ile for JA-Ile (isoleucine jasmonic acid conjugate), and OPDA for cis-OPDA (cis-(+)-12-oxo-phytodienoic acid). The concentration of each phytohormone was calculated with the help of the number of leaf samples (in mg) used and the analyte peak area with a retention time (RT) with respect to the IS peak area with IS-RT.

### 2.15. Statistical Analysis

A complete randomized design was utilized with three biological replicates for each treatment. For comparing the means of distinct replicates, an individual Student's *t*-test and Tukey's studentized range test were employed using SAS software (version 9.1, Cary, NC, USA) (model PROC GLM DATA = mydata). Except as otherwise specified, the results were predicted for variations between the means with a significance level of $p < 0.05$.

## 3. Results

### 3.1. Morphological Variations

Throughout the results of this study, the control groups will be represented as C1–C8 on day 5 and C9–C16 on day 10, and the drought-stressed (treatment) groups will be represented as T1-T8 during day 5, while the re-irrigated (treatment) groups will be represented as R1–R8 during day 10, with respect to all four photo-periodic time periods (both day 5 and day 10) and two spinach genotypes, Malav Jyoti (treatments with even numbers) and Delhi Green (treatments with odd numbers). From this, Figure 1 is excluded, as the results are not on the basis of circadian intervals. Therefore, in that case, the controls were C1 and C2 on day 5 and C3 and C4 on day 10; T1 and T2 were for the stressed plants; while R1 and R2 were for the re-irrigated plants for both genotypes, respectively.

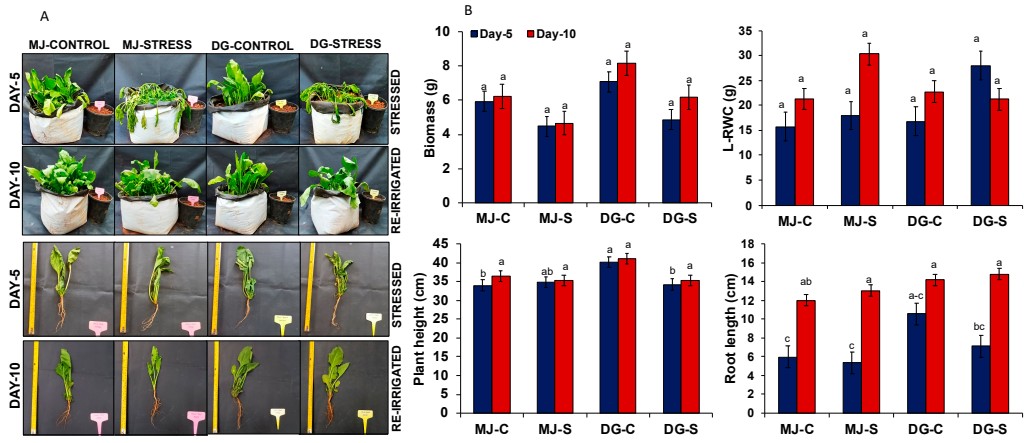

**Figure 1.** (**A**) The first two rows indicate the morphological appearance of spinach genotypes Malav Jyoti (MJ) and Delhi Green (DG), and the following two rows indicate the morphological difference in plant height, with provided proof for the leaf and root morphology. (**B**) Growth parameters (biomass, plant height, and root length) and analysis of the osmotic balance by leaf-relative water content (L-RWC). The vertical bars indicate the mean ± S.E. for n = 3 (biological triplicates). Means with different letters are significantly different at $p \leq 0.05$ according to Tukey's studentized range. The blue color bars indicate the day-5 samples (drought-stressed along with the control groups), and the red color bars indicate the day-10 samples (re-irrigated along with the control groups).

#### 3.1.1. Plant Physiology and Morphology

It was observed that the drought-stressed plants and re-irrigated plants showed a significant decrease in height when compared to the control plants, except in MJ during day 5; both the control group and drought-stressed plants had no large difference in plant height. This indicates that T2, R1, and R2 have shown a substantial variance, whereas T1 had no significant difference in plant height (Figure 1B). The heights of the spinach plants were analyzed using the three biological replicates for each group of samples; for instance, the three various plant samples were collected from three different grow bags. In both MJ and DG, during day 5 of drought stress, the control plants appeared to be healthier and taller, contrasted with drought-treated spinach plants, which were weak and wilted. However, on day 10 of the re-irrigated condition, the plants were still healthy and tall in

the control groups, whereas the re-watered plants were partially healthy and not wilted. The stem area showed a decrease in T2 compared to T1 of the drought-stressed plants; also, hypocotyl length showed an increase with respect to the drought-stressed plants after re-watering; that is, R1 and R2 had an increased hypocotyl length compared to T2 and T1 (Figure 1A).

### 3.1.2. Leaf Morphology and Structural Traits

The leaf samples of MJ and DG of the control groups appeared broad and dark green in color on days 5 and 10. In contrast, the leaves that underwent drought stress were shrunken and light green or yellowish-green or yellowish-brown in color, along with a decrease in leaf area, whereas the leaves of re-watered plants were light green in color, and few were slightly wilted. There was a moderate decrease in the total leaf area (due to a reduction in the width area) and the number of leaves observed for the plants under drought stress. Severe leaf rolling was also observed for the plants under drought stress (T1, T2) compared to all the control plants and re-watered plants (Figure 1A).

### 3.1.3. Root Physiology and Morphology

The root lengths of the spinach plants grown under drought stress reduced significantly for both MJ and DG genotypes. In contrast, both MJ and DG genotypes increased significantly in root lengths. In comparison to the drought-treated plants during day 5, the T2 plants had a significant decrease in root lengths compared to the T1 plants, which remained almost constant to C1. However, on day 10, the R1 plants showed a significant increase when compared to the R2 plants (they remained constant in the C4 group) and C3 group (Figure 1B). As spinach is known to be a dicot plant, the roots were found to appear in a tap root form with a set of secondary structures of the root, which was clearly visible in the control group samples (C1, C2, C3, and C4) of both days 5 and 10; whereas, the drought-stressed samples (T1 and T2) and plants under re-watered conditions (R1 and R2) appeared to have tap roots along with secondary and tertiary root systems (Figure 1A). Similar to the plant height, the length of the roots was measured by having three various plants from three different grow bags (biological replicates).

### 3.2. Plant Biomass and Leaf Relative Water Content (L-RWC)

Plant biomass showed a significant reduction in the case of both days 5 and 10. At day 5, T1 and T2 showed a significant decrease compared to C1 and C2; similarly, R1 and R2 showed a significant decrease compared to C3 and C4. There was a slight variation in the range of results observed; for instance, on day 10, all conditions (C3, C4, R1, and R2) had a slight increase compared to the C1, C2, T1, and T2 plant samples, respectively (Figure 1B). Considering the leaf-relative water content (L-RWC), at day 5, there was a significant increase noticed in the case of T1 when compared to C1 under drought stress; similarly, T2 showed a significant increase than that of C2. On day 10, R1 exhibited a considerable increase compared to C3, whereas R2 showed a decrease compared to C4 (Figure 1B).

### 3.3. Photosynthetic Measurements

The photosynthetic variables, such as NPR, SC, and TR, have shown similar results with a significant decrease under drought stress (T1–T8) compared to the control group (C1–C8) during day 5. Whereas, by day 10, after re-watering, the samples (R2, R5–R8) showed a significant rise in NPR, SC, and TR compared to all control group samples (C9–C16), except for the R1, R3, and R4 samples which showed a significant reduction in NPR, SC, and TR (Figure 2A–C). The chlorophyll fluorescence with the unit of Fv/Fm showed a significant increase for all drought-stressed groups (T1–T8) by day 5 when

compared to the control group samples (C1–C8). On day 10, R1 and R5 showed a significant increase compared to their controls (C9 and C13), and R7 also increased with no significance compared to the control (C15). Moreover, R2, R3, R4, R6, and R8 showed a significant reduction compared to the control group samples (C10, C11, C12, C14, and C16) (Figure 2D).

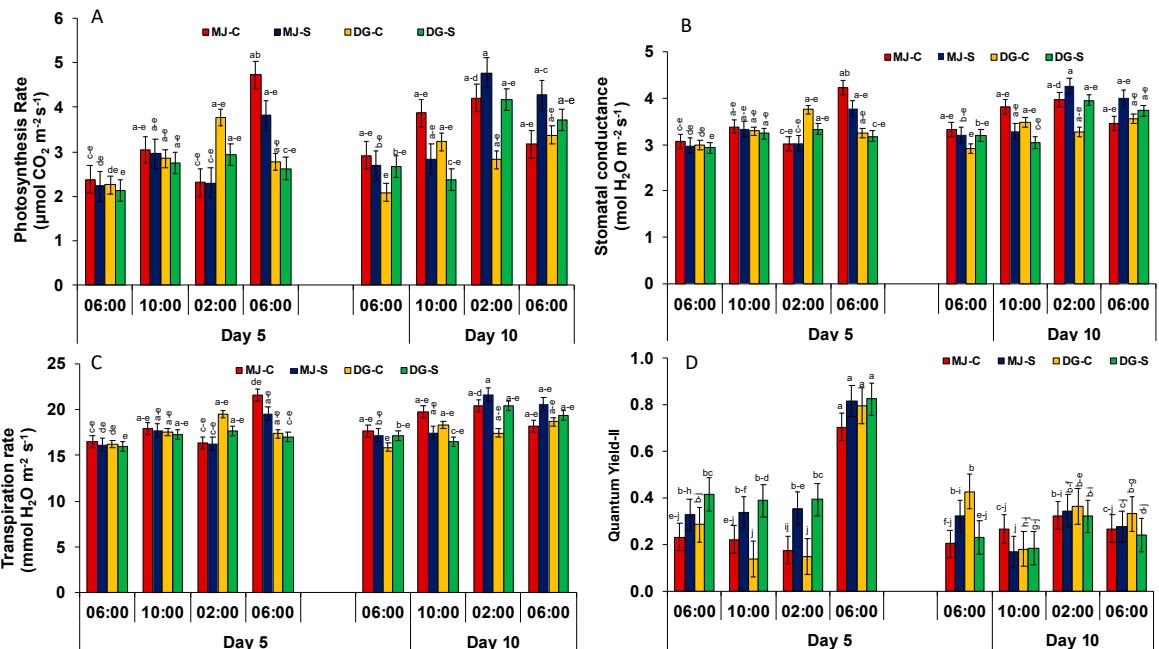

**Figure 2.** Gas exchange parameters: (**A**) Net photosynthetic rate; (**B**) stomatal conductance; (**C**) transpiration rate; (**D**) and chlorophyll fluorescence. The vertical bars indicate the mean ± S.E. for n = 3 (biological triplicates). Means with different letters are significantly different at $p \leq 0.05$ according to the Student's *t*-test. The red color bar indicates the MJ−C sample, the blue color bar indicates the MJ−S sample, the yellow color bar indicates the DG−C sample, and the green color bar indicates the DG-S during circadian time intervals of 4 h, which were exactly at 6 and 10 a.m. and 2 and 6 p.m. at day 5 and day 10.

### 3.4. Chlorophyll Molecules

The total chlorophyll and carotenoid (photosynthetic pigments or chlorophyll molecules) contents showed almost a similar result during both days 5 and 10. Considering the total chlorophyll content at day 5, apart from T7 (shown significant reduction), all other drought-stressed samples (T1–T6 and T8) showed a significant rise in the chlorophyll content when compared to the respective control group samples (C1–C8). Contrarily, after re-watering on day 10, the R2, R5, and R7 samples showed a significant drop in the total chlorophyll content when compared to their control group samples (C10, C13, and C15). The other set of plants under re-irrigation (R1, R3, R4, R6, and R8) showed a significant increase in their control samples (Figure 3A). However, regarding the carotenoid content on day 5, T5, T6, and T7 showed a significant decrease in the carotenoid content compared to other drought-stressed samples and their control group samples. Whereas, by day 10 after re-watering, the R1, R2, R4, R5, R7, and R8 samples remained constant in their respective control group samples. In contrast, R3 and R6 showed a significant rise in the carotenoid content compared to their control samples (C11 and C14) (Figure 3A).

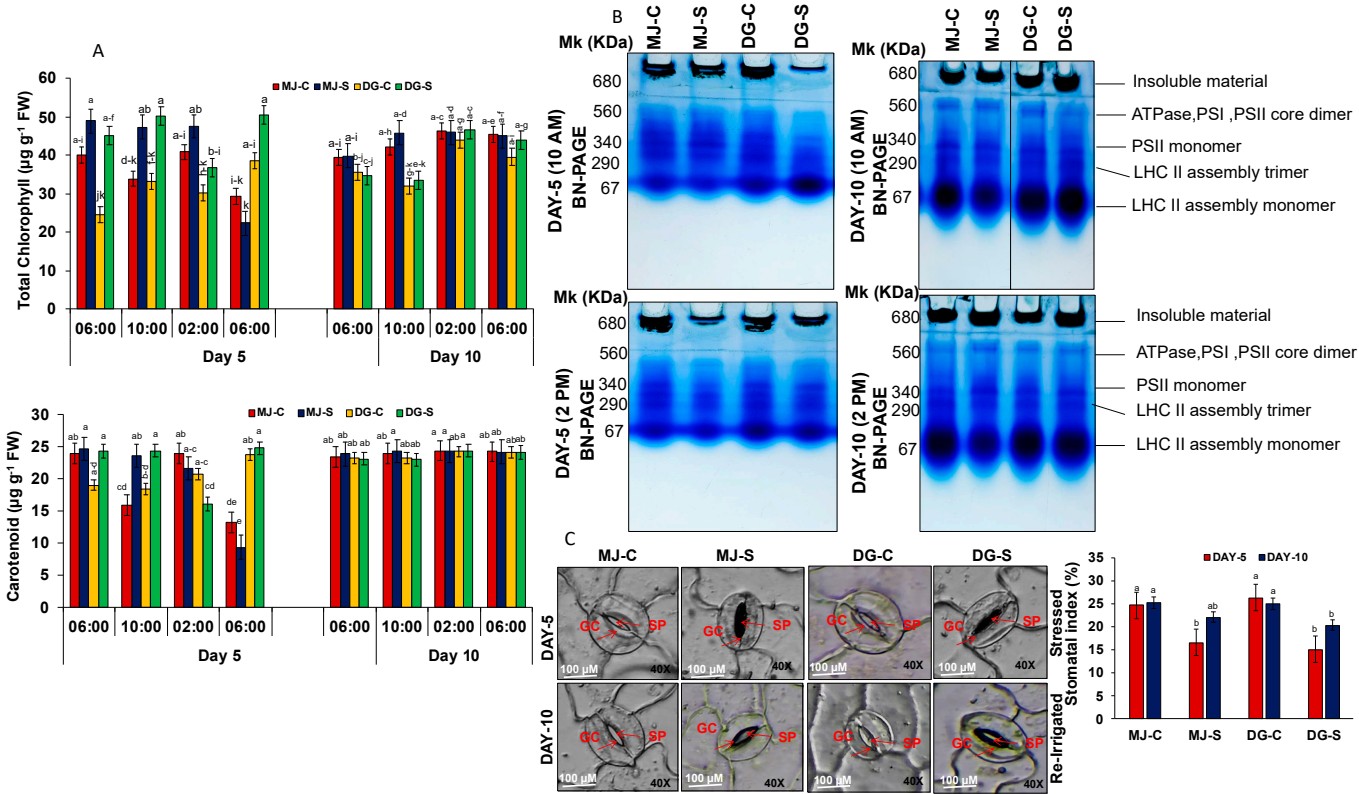

**Figure 3.** (**A**) Chlorophyll pigments (total chlorophyll content and carotenoid content) at 4 h circadian time interval; (**B**) thylakoidal proteome by BN-PAGE technique was performed for samples collected at 10 a.m. and 2 p.m. on day 5 and day 10 and represents the photosynthetic proteins present in spinach genotypes; (**C**) structure of stomata and stomatal index on day 5 and day 10. The vertical bars indicate the mean ± S.E. for n = 3 (biological triplicates). Means with different letters are significantly different at $p \leq 0.05$ according to Tukey's studentized range (for stomatal index) and Student's *t*-test (for chlorophyll pigments). The blue color bars indicate the day-5 samples, and the red color bars indicate the day-10 samples for stomatal index.

### 3.5. Thylakoidal Protein Analysis (BN-PAGE)

Blue Native Poly-Acrylamide Gel Electrophoresis (BN-PAGE) was carried out to segregate the multi-protein complex proteins (MCPs) from the thylakoidal region in spinach genotypes for 10 a.m. and 2 p.m. on both day 5 (drought stressed) and day 10 (re-irrigated). The spinach leaf samples were collected in three biological replicates. In Figure 3B, the gels consisting of the native protein profile of thylakoidal MPCs were extracted on day 5 (10 a.m. and 2 p.m.) from the control (MJ-C and DG-C) and drought-stressed samples (MJ-S and DG-S). On the other hand, on day 10, we used the re-irrigated (MJ-S and DG-S) and control samples (MJ-C and DG-C). The protein band that has shown expression in the gel at the range of 680–560 KDa was ATPase, PS-I, and PS-II core dimer. These proteins were downregulated in case of drought stress and were upregulated in case of re-irrigation compared with the control samples during all four circadian hours. The consecutive next band appeared in the range of 340 KDa, and the protein present was PS-II monomer (or) Cyt b6/f. The final two different bands contained proteins, namely the LHC-II assembly trimer and LHC-II assembly monomer at the ranges of 290 and 67 KDa, respectively.

### 3.6. Stomatal Index and Stomatal Appearance

As proof of the results observed from the cellular experiments, the stomatal structures were visualized with the help of micrographs on days 5 and 10. On day 5, the stomata of the control samples (C1 and C2) had the stomatal pore (SP) open, and the guard cells

(GCs) appeared turgid, whereas the drought-stressed samples (T1 and T2) had closed SPs and GCs and were fully turgid. The chloroplast had plenty of smaller granules in the case of the stressed group; the control group had the chloroplast intact compared to the drought-stressed group. However, on day 10, the re-irrigated plant samples (R1 and R2) had the SP partially open, and the GCs were partially turgid, whereas their control samples (C3 and C4) had the SP open and the GCs were not turgid (Figure 3C).

The results obtained from the stomatal index have shown a significant decrease in the case of both the drought-stressed group (T1 and T2) and re-irrigated plant samples (R1 and R2) when compared to their control groups (C1, C2 and C3, C4), during days 5 and 10, respectively. The control groups at days 5 and 10 had no significant difference, whereas the re-irrigated plant samples (R1 and R2) had an increase in the stomatal index compared to the drought-stressed samples (T1 and T2). The stomatal index in percentage was derived using the number of stomata and epidermal cells present in a unit area of the leaf samples (Figure 3C).

### 3.7. Malonaldehyde (MDA) and Proline Content

On day 5, from 6 am to 6 p.m., it was noticed that the drought-stressed samples (T1–T8) showed a significant increase when compared to their respective control samples (C1–C8). From 6 am, it showed an increase, and gradually, it reached the highest increase by 2 p.m. (mainly T5); further, by 6 p.m., it was less when compared to the 2 p.m. samples but higher than the 10 a.m. samples. However, on day 10, the MDA content was less in comparison with day 5; after re-watering, the DG samples (R2, R4, R6, and R8) decreased significantly with MDA content when compared to the control samples (C10, C12, C14, and C16). The MJ re-watered samples (R1, R3, R5, and R7) by day 10 in all photo-periodic hours showed a significant rise in MDA content compared to their control samples (C9, C11, C13, and C15) and DG re-watered plant samples (Figure 4A). The proline content showed a significant increase under drought stress and re-irrigation at days 5 and 10. Apart from T4 (which remained constant to its control) and T7 on day 5 and R4, R5, and R7 on day 10, all other conditions showed an increase in proline content significantly, whereas the mentioned treatments showed a significant drop in proline content compared to the respective control (untreated) samples (Figure 4B).

### 3.8. $H_2O_2$ and $O_2^-$ Localizations

Increased ROS accumulation causes a rise in the MDA (or LPO) content in the form of $H_2O_2$ and $O_2^-$. To depict the results (oxidative stress) at the cellular level, two major biochemical analyses with the help of staining methods (DAB and NBT) were conducted, as we already noticed the MDA and proline content. The DAB ($H_2O_2$) staining (Figure 4C) produced the observed results of dark brown color spots on leaf samples, whereas dark blue color spots (formazon formation) occurred on leaves in the case of the NBT ($O_2^-$) staining (Figure 4D) test. The stressed groups (T3, T4, T5, and T6) had spots on the inner layer and the color change on the outer layer of leaves by day 5 (10 a.m. and 2 p.m.), whereas the re-watered groups (R3, R4, R5, and R6) had a reduction in spots and the outer layer color changed partially at day 10 (10 a.m. and 2 p.m.). The control groups (T3, T4, T5, and T6) had no change in color on the outer layer of the leaves, and the spots were also comparatively less on day 5, whereas, on day 10, by 2 p.m., the controls (C13 and C14) had reduced spots, which were more notable for both DAB and NBT. This has shown a reduction in ROS accumulation in spinach genotypes (MJ and DG) with respect to circadian hours after re-irrigation.

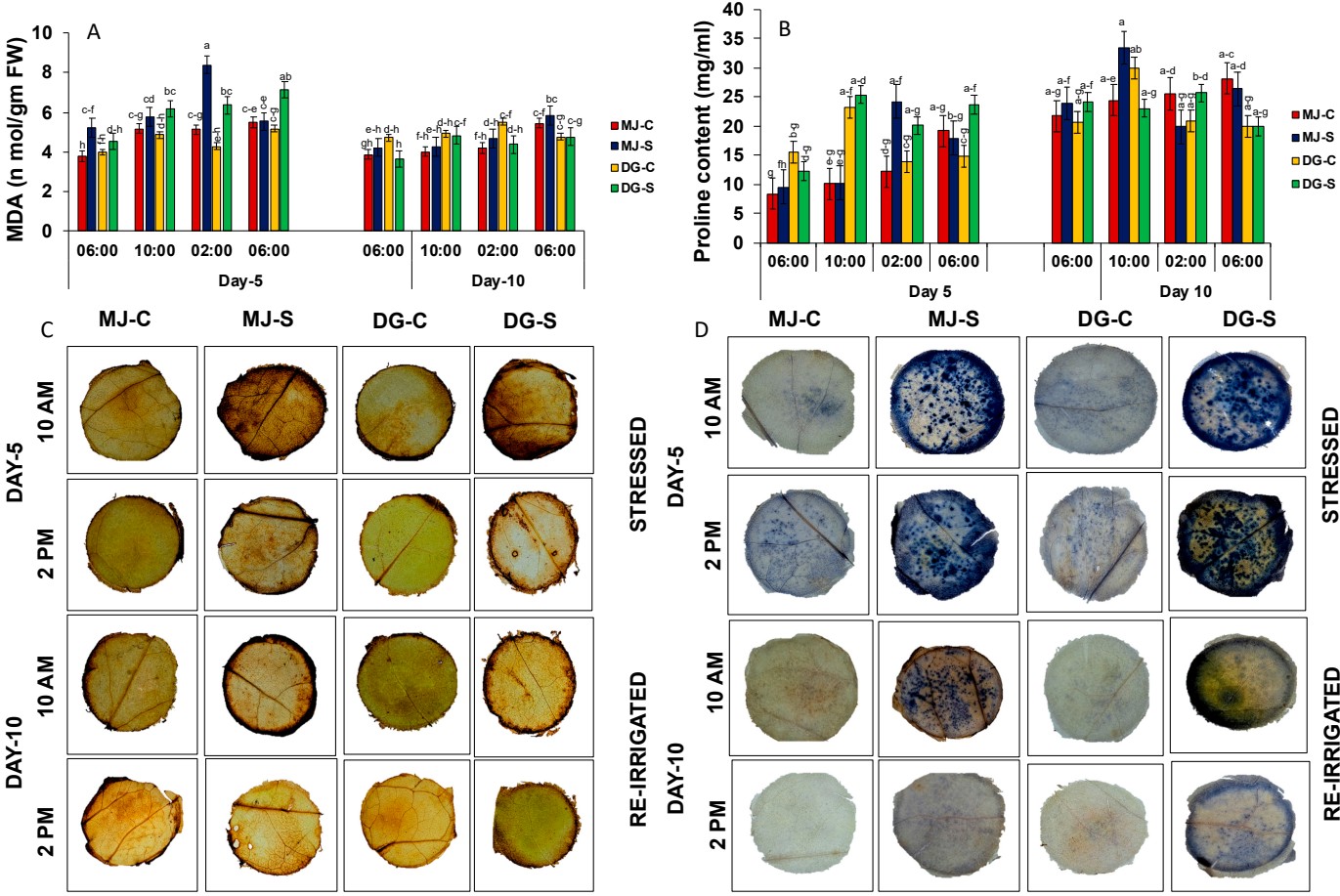

**Figure 4.** Oxidative stress markers: (**A**) MDA content by TBARS method; (**C**) $H_2O_2$ localization by DAB staining method; (**D**) $O_2^-$ localization by NBT staining method and the osmoprotectant; and (**B**) proline content by ninhydrin method. The vertical bars indicate the mean ± S.E. for n = 3 (biological triplicates). Means with different letters are significantly different at $p \leq 0.05$ according to Tukey's studentized range. DAB and NBT staining was performed at 10 a.m. and 2 p.m. Spinach samples were collected on day 5 and day 10.

*3.9. Total Soluble Protein (TSP) and Total Phenolic Content (TPC)*

The TSP for all drought-stressed samples and re-irrigated samples had increased significantly at both day 5 and day 10 when compared to their control samples. The T8 and R6 samples showed the highest increase in TSP, which was DG-S by 6 p.m. on day 5 and DG-S by 2 p.m. on day 10. Apart from the 2 p.m. samples on day 5 and 6 p.m. samples on day 10, all other circadian hour samples showed a significantly increased value with TSP in both spinach genotypes (Figure 5A). The TPC of all drought-stressed samples associated with circadian hours by day 5 showed a significant increase in TPC except for the T8 (DG-S of 6 p.m.) sample when compared to their control samples (Figure 5B). Whereas, by day 10, the TSP showed a significant decrease in TPC after re-watering the spinach samples except R3 (MJ-S of 10 a.m.), when compared to the control samples. The TPC was at its highest increase between 10 a.m. and 6 p.m. on day 5, mainly T3 and T7 (MJ plants).

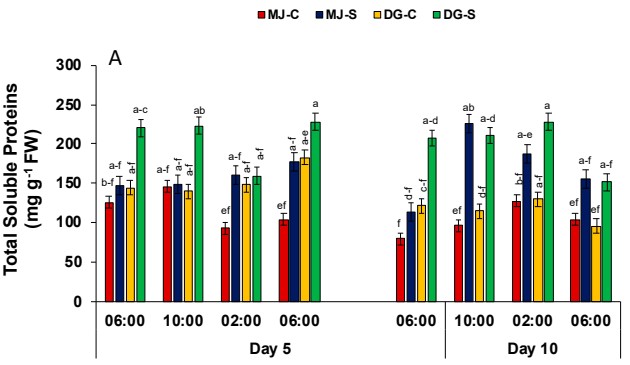
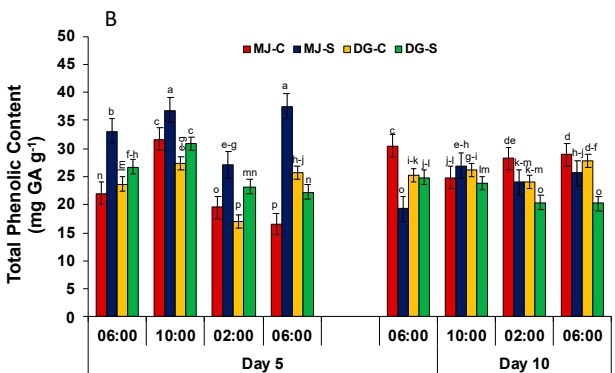

**Figure 5.** (**A**) Total soluble proteins using the Bradford method (TSP) and non-enzymatic antioxidant; (**B**) total phenolic content (TPC) using the Folin–Ciocalteau method for the spinach samples collected during circadian time intervals of 4 h, which was exactly at 6 and 10 a.m. and 2 and 6 p.m. on day 5 and day 10. The vertical bars indicate the mean ± S.E. for n = 3 (biological triplicates). Means with different letters are significantly different at $p \leq 0.05$ according to Tukey's studentized range (for TSP) and Student's $t$-test (for TPC).

### 3.10. Antioxidant Enzymatic Activities and Their Native PAGE Profiling

The formation of ROS was confirmed through the above results of $H_2O_2$ and $O_2^-$ localization with drought stress and re-irrigated spinach leaf samples by day 5 and day 10. To investigate the detoxification of oxidative damage, three crucial enzyme activities, along with their respective isozymes were estimated. The activities of these antioxidant enzymes, which play a vital role in the homeostasis of ROS in plants, such as superoxide dismutase (SOD), catalase (CAT), and ascorbate peroxidase (APX), were altered distinctly under drought-stressed conditions. The results observed for SOD activity showed a significant increase in the case of T1 and T4 under drought, whereas T3 and T8 remained constant under drought stress with their control samples (C1–C8) on day 5. Apart from that, other treatments (T2, T5–T7, and R1-R8) decreased significantly in SOD activity in comparison with all their control group plants on both day 5 and day 10 (Figure 6A). The SOD activity was validated with the isozyme expression patterns depicted with three different isoforms during 10 a.m. and 2 p.m. on both day 5 and day 10. The band intensities of the SOD-2 and SOD-3 isoforms were expressed as high compared to SOD-1 during all four circadian hours, mainly in the case of the drought and re-irrigated samples (Figure 6B).

Considering CAT activity, on day 5, the drought-stressed samples (T1–T6, T8) showed a significant increase compared to the control (C1–C6, C8) samples at all circadian hours, except T7, which had a contrary result with a significant reduction in CAT activity when compared to the control (C7) sample. Whereas on day 10, after re-watering, those samples (R1–R8) showed a significant decrease in CAT activity for all circadian hours (at 6 and 10 a.m. and 2 and 6 p.m.) when compared to their control (C9–C16) samples (Figure 6A). The CAT activity was validated with the proof of isozyme expression patterns, which depicted the two different isoforms at 10 a.m. and 2 p.m. on both day 5 and day 10. The band intensity of the CAT-2 isoform had a higher expression compared to CAT-1, particularly for the drought sample at day 5 (10 a.m. and 2 p.m.) (Figure 6B). The APX activity also showed a significant increase in the case of all drought-stressed samples (T1–T3 and T5–T8) on day 5, except T4, which showed a contrasting result of significant reduction compared to the control (C4) sample. On day 10, after re-irrigation, those plant samples (R1-R8) reduced significantly in APX activity when compared to their control (C9–C16) samples (Figure 6A). The APX activity was validated using isozyme expression patterns and also depicted the three different isoforms at 10 a.m. and 2 p.m. on both day 5 and day 10. The band intensities of APX-1 and APX-2 were noticed to be higher in expression when

compared to APX-3 for both the drought and re-irrigated samples on day 5 and day 10, respectively (Figure 6B).

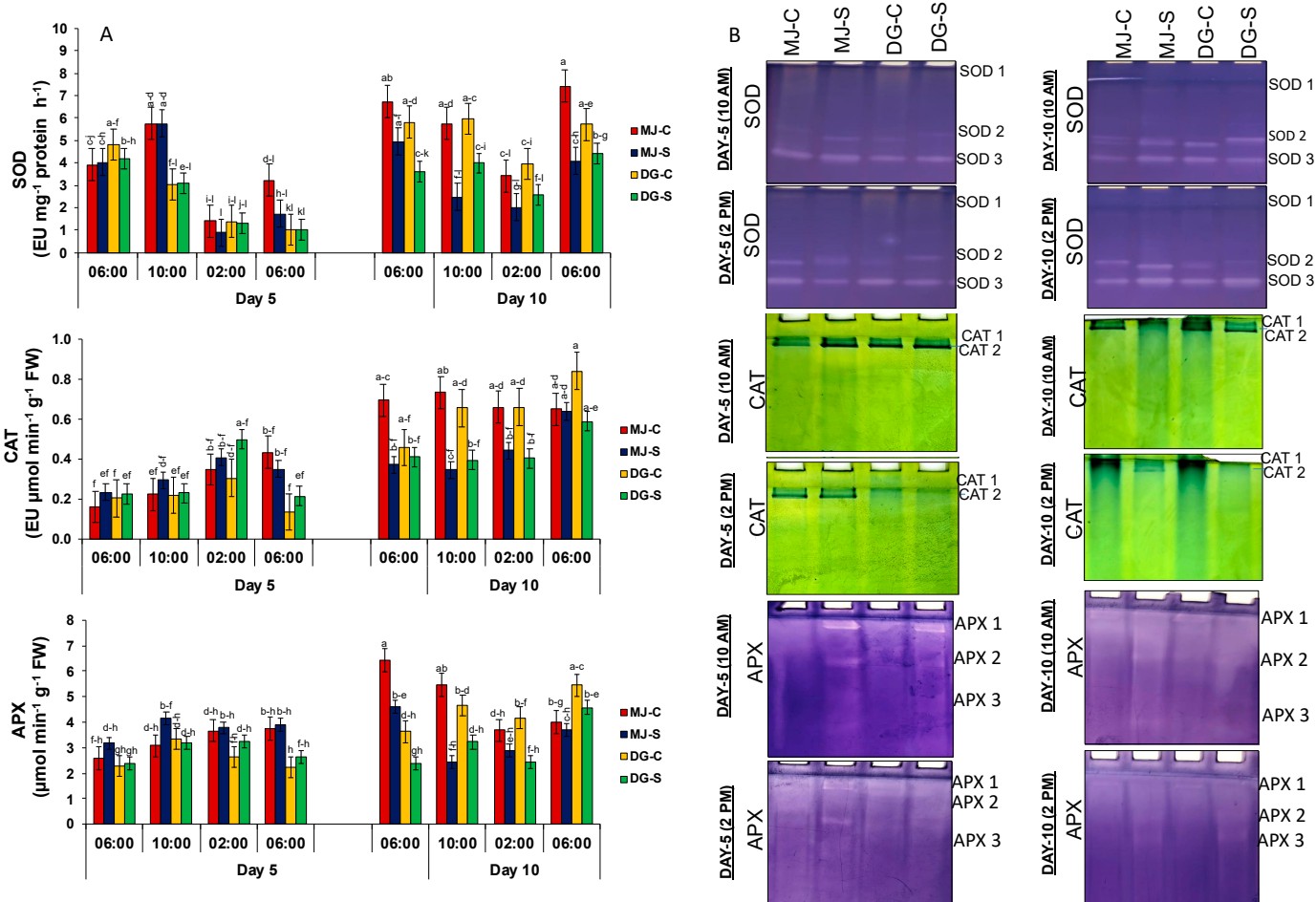

**Figure 6.** Enzymatic antioxidants: (**A**) SOD CAT and APX; the red color bar indicates the MJ–C sample, the blue color bar indicates the MJ-S sample, the yellow color bar indicates the DG–C sample, and the green color bar indicates the DG–S during circadian time intervals of 4 h, which was exactly at 6 and 10 a.m. and 2 and 6 p.m. on day 5 and day 10. (**B**) Their isozymes by the Native–PAGE method were performed at 10 a.m. and 2 p.m.; spinach samples were collected on day 5 and day 10. The vertical bars indicate the mean ± S.E. for n = 3 (biological triplicates). Means with different letters are significantly different at $p \leq 0.05$ according to Tukey's studentized range (for CAT) and Student's *t*-test (for SOD and APX).

### 3.11. Quantitative RT-PCR Analysis

To validate our findings and to understand the circadian system associated with drought stress in spinach samples, we orchestrated a supportive experiment employing quantitative real-time PCR (qRT-PCR). The gene selection was conducted with respect to the circadian hours (morning–evening loop) and drought stress with biological triplicates. The selected circadian genes were *CCA1*, *LHY*, *RVE8*, *PRR5*, *PRR7*, *TOC1*, *PRR3*, and *PRR9*, and the drought-responsive genes were *DREB1*, *DREB2*, and *PIP1*.

#### 3.11.1. Expression of Core Genes of Circadian Clock

The qRT-PCR was performed for the 10 a.m. and 2 p.m. samples on both day 5 and day 10. Considering *CCA1*, the expression of re-watered plant samples (R3 and R4) on day 10 showed a significant increase compared to the control samples (C11 and C12), whereas the expression of R5 and R6 showed a significant reduction in expression compared to their

control samples (C13 and C14). However, on day 5, under drought stress, we observed a contrasting result with downregulation for T3 and T4 compared to the control samples (C3 and C4). Whereas, by 2 p.m. on day 5, the drought-stressed samples (T5 and T6) showed a significant rise in expression compared to the control samples C5 and C6 (Figure 7). The expression of *LHY* appeared almost similar to the expression of *CCA1*. The *LHY* gene expressed an increase at 2 p.m. on day 5 with the T5 sample; similarly, there was a notable increase observed for the T6 samples, whereas T3 showed a significant reduction compared to the control sample, and T4 remained constant with their control (C4) samples. However, by day 10 after re-irrigation, between 10 a.m. and 2 p.m., the R3 and R4 samples had increased significantly in *LHY* gene expression compared to their control samples C11 and C12. In contrast, R5 and R6 (2 p.m. during day 10) showed a significant reduction compared to the control (C13 and C14) samples (Figure 7).

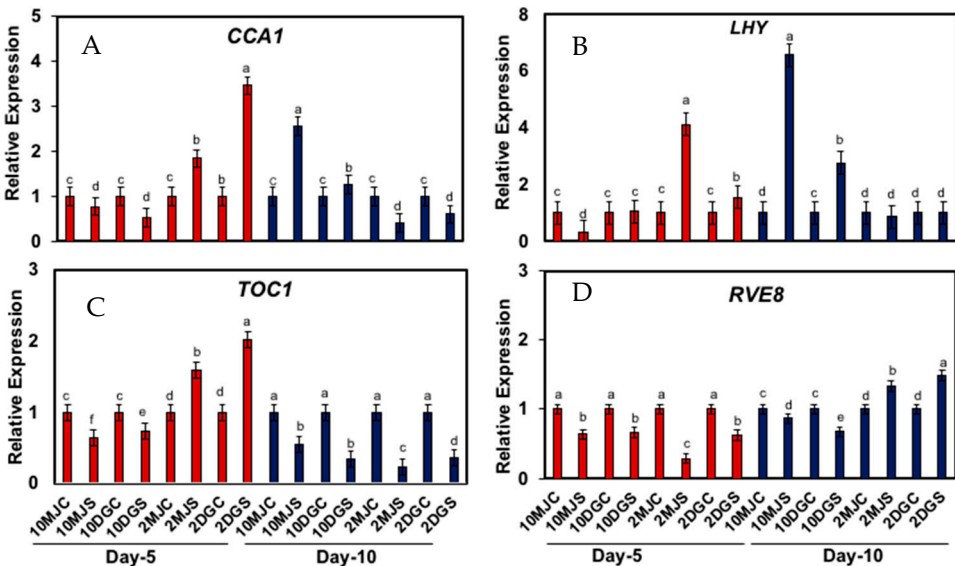

**Figure 7.** Relative expression or fold change in circadian core genes (**A**) *CCA1*, (**B**) *LHY*, (**C**) *TOC1*, or *PRR1* and (**D**) *RVE8*. The results are represented for spinach genotypes MJ and DG at 10 a.m. and 2 p.m. that were collected on day 5 (drought stressed along with the control groups) and day 10 (re-irrigated along with the control groups). The vertical bars indicate the mean $\pm$ S.E. for n = 3 (biological triplicates). Means with different letters are significantly different at $p \leq 0.05$ according to Tukey's studentized range.

Regarding TOC1, under drought-stressed conditions on day 5, the stressed samples (T3 and T4) at 10 a.m. decreased significantly in expression compared to the untreated or control samples (C3 and C4), whereas by 2 p.m., the stressed samples (T5 and T6) were increased in expression when compared to their control (C5 and C6) samples. However, by day 10, after re-watering, all samples by both 10 a.m. (R3 and R4) and 2 p.m. (R5 and R6) showed a significant reduction compared to their control samples (C11, C12, and C13, C14), respectively (Figure 7). Regarding RVE8, during day 5 of drought stress, all stressed samples showed a significant reduction at both 10 a.m. and 2 p.m.; similarly, by day 10, the 10 a.m. re-watered samples (R3 and R4) showed a significant reduction compared to the control samples. Contrarily, by 2 p.m., the expression of re-watered samples (R5 and R6) showed a significant increase compared to their control samples (Figure 7).

### 3.11.2. Expression of Pseudo-Response Regulators (PRRs)

Considering *PRR3*, apart from the 2 p.m. drought-stressed samples (T5 and T6 showed a significant increase) on day 5, the expression of all other treatments (T3 and T4 on day 10; and R3, R4, R5, and R6 on day 10) showed a significant reduction compared to their control

samples (Figure 8). Regarding *PRR5*, the expression of the drought-stressed samples by day 5, T3 (10 a.m.), and T6 (2 p.m.) showed a significant increase, and T4 (10 a.m.) and T5 (2 p.m.) showed a significant decrease in expression compared to the control samples (C3–C6). After re-watering, by day 10, R3 and R4 at 10 a.m. showed a significant reduction in expression, whereas R5 and R6 at 2 p.m. showed a significant increase compared to their control (C11–C14) samples (Figure 8).

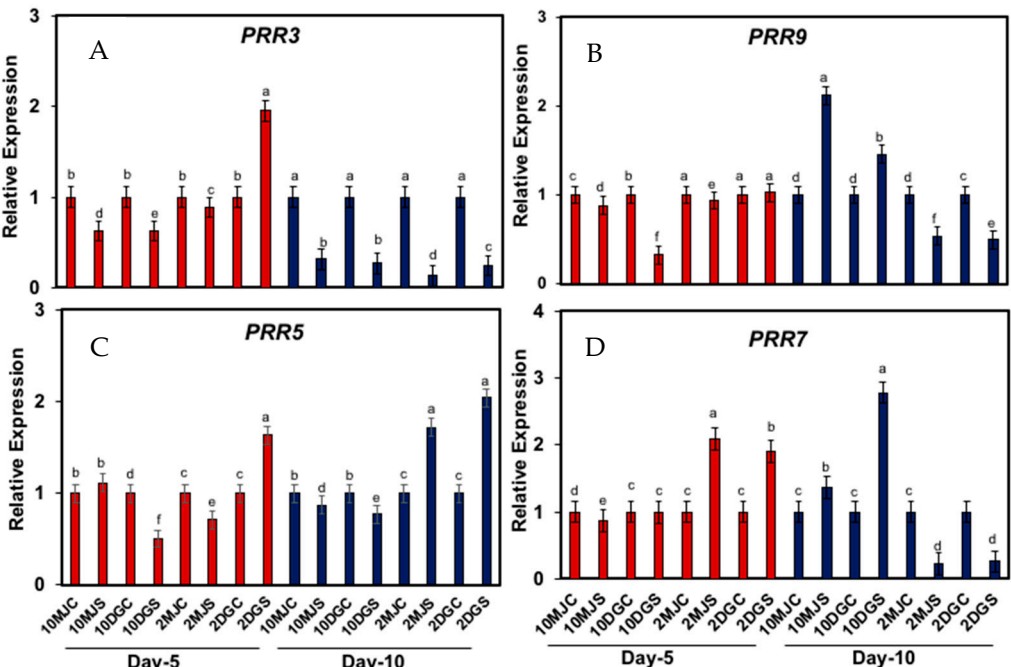

**Figure 8.** Relative expression or fold change in pseudo-response regulators present in circadian oscillator: (**A**) *PRR3*, (**B**) *PRR5*, (**C**) *PRR7*, and (**D**) *PRR9*. The results are represented for spinach genotypes MJ and DG at 10 a.m. and 2 p.m. collected on day 5 (drought stressed along with control groups) and day 10 (re-irrigated along with control groups). The vertical bars indicate the mean $\pm$ S.E. for n = 3 (biological triplicates). Means with different letters are significantly different at $p \leq 0.05$ according to Tukey's studentized range.

*PRR7* and *PRR9* are the morning-expressed genes, and here, *PRR7* for T3 (10 a.m. on day 5), R5, and R6 (2 p.m. on day 10) reduced significantly, and T4 remained constant to its control (C4) sample. In contrast, T5, T6 (2 p.m. on day 5) and R3, R4 (10 a.m. on day 10) have increased significantly with their expression of *PRR7* compared to their control samples (Figure 8). Considering the results of *PRR9*, in the treatment groups, T6, which remained constant to C6, R3, and R4, showed a significant increase in expression compared to C11 and C12. Contrarily, T3, T4, and T5 on day 5 and R5 and R6 on day 10 showed a significant decrease in expression when compared with the control samples, such as C3, C4, and C5 on day 5 and C13 and C14 on day 10, respectively (Figure 8).

3.11.3. Expression of Drought-Responsive Genes

*DREB1* and *DREB2* are drought-responsive genes that are highly expressed, mainly under drought stress. Both *DREB1* and *DREB2* have shown similar results in our study where on day 5, under drought stress, T3 (was increased at highest), T4, T5 (was increased at highest), and T6 showed a significant increase compared to C3, C4, C5, and C6. In contrast, on day 10 after re-watering, R3, R4. R5 and R6 showed a significant decrease compared to C11, C12, C13, and C14 (Figure 9). *PIP1*, by day 5 under drought stress, T3 and T4 were increased at the highest by 10 a.m.; similarly, by 2 p.m., T5 and T6 showed an upregulated expression compared to C3, C4, C5, and C6. After re-watering, by day 10, at

10 a.m., R3 and R4 remained almost constant with significance compared to C11 and C12. In contrast, by 2 p.m., R5 and R6 showed a significant decrease when compared to their control (C13 and C14) samples (Figure 9).

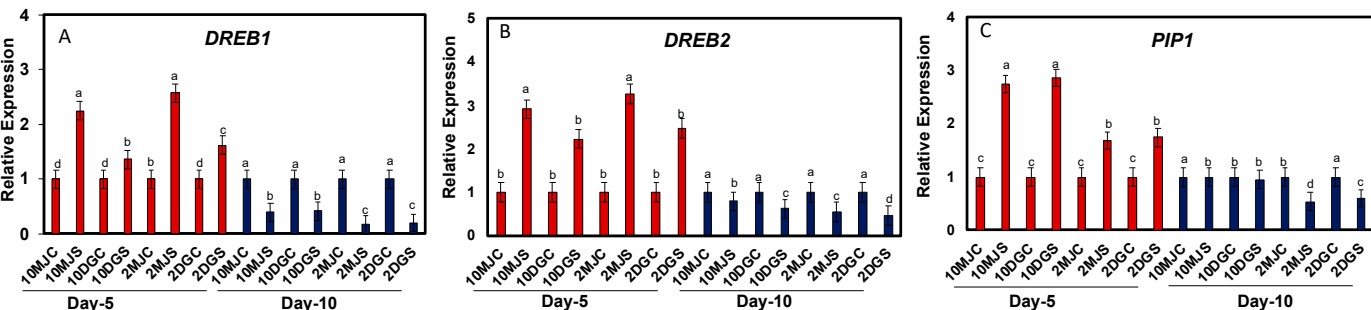

**Figure 9.** Relative expression or fold change in drought-related genes (**A**) *DREB1*, (**B**) *DREB2*, and (**C**) *PIP1*. The results are represented for spinach genotypes MJ and DG at 10 a.m. and 2 p.m. collected on day 5 (drought stressed along with control groups) and day 10 (re-irrigated along with control groups). The vertical bars indicate the mean ± S.E. for n = 3 (biological triplicates). Means with different letters are significantly different at $p \leq 0.05$ according to Tukey's studentized range.

### 3.12. Targeted Metabolomics Analysis

The concentration level of JA in spinach leaf samples at 10 a.m. and 2 p.m. on day 5 showed a significant decrease under drought stress when compared to their respective controls and re-irrigated samples. However, the highest increase in JA concentration was observed after re-irrigating the spinach samples compared to the drought samples; although the samples showed the highest increase in the re-irrigated group have yet shown a significantly reduced JA concentration compared to their control samples (Figure 10). Considering ABA, the day-5 samples under drought stress (T3–T6) showed a significant increase compared to their control samples (C3–C6). Contrarily, during day 10, after re-irrigation, R3 and R4 showed a significant reduction in ABA concentration with spinach compared to the control samples (C11 and C12), whereas R5 and R6 had a significant increase in ABA with the spinach genotypes compared to their control (C13 and C14) samples (Figure 10).

The concentration level of SA in the spinach samples on both day 5 and day 10 (10 a.m. and 2 p.m.) showed a significant reduction, except T4 and T5, which showed a significant increase in SA concentration in DG-S on day 5 (10 a.m.) and MJ-S on day 5 (2 p.m.) compared to the C4 and C5 control samples (Figure 10). Considering the concentration of JA-Ile in spinach genotypes, on day 5, T3 (MJ-S at 10 a.m.) remained constant with C3; T4, T5, and T6 showed a significant reduction in JA-Ile concentration compared to their respective control groups (C4, C5, and C6). However, on day 10, after re-irrigation, R4 and R6 (DG at 10 a.m. and 2 p.m.) showed a significant decrease in JA-Ile concentration compared to the C12 and C14 control samples. In contrast, R3 and R5 (MJ at 10 a.m. and 2 p.m.) showed a significant increase compared to their control samples C11 and C13 (Figure 10). The concentration of cis-OPDA in both spinach genotypes under drought stress on day 5 (10 a.m. and 2 p.m.) showed a significant reduction compared to their control samples. However, on day 10, after re-irrigation, R4, R5, and R6 showed a significant rise in cis-OPDA concentration compared to their (C12, C13, and C14) control samples. Contrarily, R3 showed a significant drop in cis-OPDA concentration compared to the untreated or control samples, C11 (Figure 10).

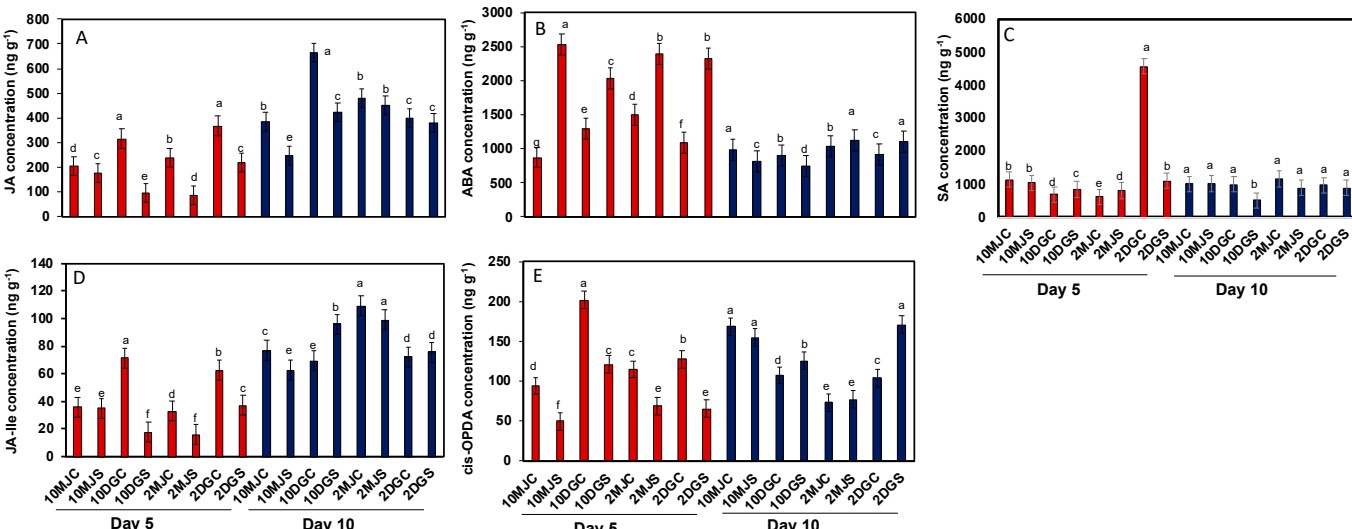

**Figure 10.** Metabolite concentration by UHPLC-MS/MS: (**A**) jasmonic acid (JA); (**B**) abscisic acid (ABA); (**C**) salicylic acid (SA); (**D**) isoleucine jasmonic acid conjugate (JA-Ile); and (**E**) cis-(+)-12-oxo-phytodienoic acid (OPDA). The results are represented for spinach genotypes MJ and DG at 10 a.m. and 2 p.m. collected on day 5 (drought stressed along with control groups) and day 10 (re-irrigated along with control groups). The vertical bars indicate the mean ± S.E. for n = 3 (biological triplicates). Means with different letters are significantly different at $p \leq 0.05$ according to Tukey's studentized range.

## 4. Discussion

Drought is one of the major factors that has a limitation on spinach growth and productivity in water-circumscribed environments [70]. It has been observed that the plant circadian clock is associated with the alleviation of abiotic stress [71]. There are few reviews (or a restricted number of research works) that have reported on how the circadian clock is associated with stress tolerance. The circadian oscillator is closely related to the adaptability of plants as it is accompanied by physiological processes depending upon the changes in environmental cues [72,73]. The circadian clock plays an essential role in acclimatizing the external environmental stresses, mainly drought stress [74]. Consequently, the current study has mainly focused on exploring three major factors: (i) How the spinach plants (MJ and DG) tolerate or combat drought stress by employing the internal circadian oscillator, provided with molecular-level analysis, including chloroplast proteome, particularly during the morning–evening loop. (ii) The current study has concentrated on providing knowledge about how the circadian clock gene and drought-responsive genes are regulated during photo-periodic hours (10 a.m. and 2 p.m.) on day 5 and day 10. (iii) We have also conducted a study on targeted metabolomics analysis with phytohormone concentrations in spinach genotypes, especially defense hormones such as ABA, JA, SA, JA-Ile, and cis-OPDA under both water regimes (drought stress and after re-irrigation).

The apparent symptoms of drought stress during the vegetative stage are a reduction in plant height, wilting of leaves, and changes in the number and area of the leaves [4]. The leaves of plants acquire smaller leaf areas, higher leaf tissue density, and larger leaf thicknesses to reconcile drought conditions [75]. The roots of the plants also play a vital role in drought stress [76]; here, spinach mainly uptakes the macro-nutrients present in the soil, helping the plants survive by absorbing and utilizing the water stored in the soil [77]. The plant height is grievously affected by drought, and it is correlated to leaf senescence and cell enlargement [4]. Similar to our results, there was a significant reduction in plant height noticed in maize hybrids [78], lily [79], rice [80], and sugarcane [81] under drought stress. The decreased plant height and biomass were observed in durum wheat under drought

conditions [82]; these results were noticed to be similar to our results, which are presented in Figure 1A,B. The reduction in plant height is particularly due to a reduction in cell expansion, impaired mitosis, and a rise in leaf shedding under drought stress. Aside from the plant height, leaves are major organs for transpiration and assimilation in plants [4]. In our study, there was an increase in RWC, mostly resulting in high osmotic regulation or lower elasticity of the tissue cell wall [83]. Similar to our results (Figure 1A,B), Zokaee-Khosroshahi [84] has identified a significant decrease in growth parameters, namely the dry and fresh weights, number of leaves, and total leaf area by using five Iranian almond species (*Prunus dulcis*, *P. haussknechti*, *P. eburnean*, *P. scoparia*, and *P. eleagnifolia*) under polyethylene glycol (PEG)-induced drought stress. In a recent study, there was a reduction observed in plant height, root length, and the dry and fresh weights of two canola cultivars when it was treated with 300 g/L of PEG drought stress [85].

Photosynthesis is one of the essential processes that is affected by drought stress. Leaf photosynthetic products are known to be a foundation material for plant growth. The reason behind the reduction in NPR and transpiration rate is the decline in soil RWC. Plant photosynthesis and yield are directly affected due to the changes occurring in leaf area, which is nearly an obvious feature that can be observed from plant leaves under drought stress [4]. The photosynthetic rate decreases due to the closure of the stomata, disturbances in certain enzymatic activity (especially those involved in ATP synthesis), and membrane damage [15]. In our study, we noticed a reduction in the photosynthetic parameters under drought stress (Figure 2); however, during day 10 after re-irrigation, 10 a.m. and 2 p.m. (Figure 2) showed an increase in photosynthetic parameters, which indicates that these photo-periodic hours in circadian biology have helped the spinach genotypes in combating the effect of drought stress. The prime reason behind the decline in NPR under drought stress is stomatal limitation and non-stomatal limitation. Under mild drought, stomatal limitation is the major reason, whereas during severe drought conditions, non-stomatal factors are the principal reason behind the decrease in NPR [86].

Thus, chlorophyll is subsequently metabolized in plants and is correlated to photosynthesis. The decreased chlorophyll pigments (green pigments) under drought stress cause differences in photosynthetic function [87]. Our results (Figure 3A) are partially similar (we observed a reduction in carotenoid content under drought stress) to Wu et al. [88], where it was found that the total chlorophyll content and carotenoids for Chinese cork oak (*Quercus variabilis*) seedlings have shown a significant reduction under various drought intensities (40% and 20%). However, it has been observed that not all plants show a decreased chlorophyll content, and also, has pointed out the increase in chlorophyll content in borage leaves under drought stress, which was stated mainly due to a lower leaf area index and high radiation interception [89]. We have also observed an increased chlorophyll content in our results, presented in Figure 3A. The alteration in chlorophyll pigments often leads to a change in the color of the plant to yellowish-brown under drought-stress conditions [4]. Also, our study focused on chloroplast proteome to understand the effect of drought stress in spinach genotypes on day 5 (10 a.m. and 2 p.m.), presented in Figure 3B; alongside this, our study has shown how exactly the circadian pattern controls the harmful effects of drought stress with the re-irrigated spinach samples on day 10 (10 a.m. and 2 p.m.).

Due to the induction of drought stress, plants undergo a set of secondary stressors, such as osmotic and oxidative stress. As MDA and proline are known to be stress indicators, this current study has initially focused on knowing how the circadian oscillator helps spinach combat drought stress. Similar to our results with certain samples, there was a decrease in MDA and proline contents observed in leucaena seedlings under drought stress [90]. In this study, we observed an increased MDA content (Figure 4A) with most of the stressed samples, as drought stress leads to an increase in MDA content, which is a LPO (lipid

peroxidation) indicator; due to the increase in LPO, the plant undergoes membrane damage, which results in a reduction in growth, which shows an increase in MDA (bi-product of LPO) content. Plants have the ability to withstand dehydration (plants do this by involving their defense systems to reciprocate oxidative stress) using various physiological activities, namely osmotic adjustment through osmoprotectants (or osmolytes) such as amino acids (betaine and proline), sugars, polyols, organic, inorganic ions, and other amino acids [91,92]. These osmolytes play an important role in maintaining the plant's cellular functions under drought stress [15,87]. The osmolyte or osmotic regulating substance proline is also known as a type of free radical scavenger [93], which is readily stored in the vacuoles of plants [94]. In this study, we have observed an increase in proline content (Figure 4B) with an increase in proline accumulation, as proline has a vital role in reducing the deleterious effects of drought and helps the plants with their stress tolerance by safeguarding enzyme protein structures and membranes [95].

In an extreme stress condition, it leads to an increase in ROS concentrations in cells and causes oxidative damage to membranes (LPO), proteins, DNA, and RNA molecules. Thus, it further leads to oxidative destruction at a cellular level, and it is known to be oxidative stress [96]. ROS metabolism is foremost important with respect to the maintenance of the oxidative physiology of plants [10]. In green plants, chloroplasts are the major site for the production of ROS [97]. The production and accumulation of ROS in spinach genotypes have been observed in our study, and it is presented in Figure 4C,D. In a previous study, it has been reported that drought stress has the ability to increase ROS, along with a severe or partial rise in oxidation of cellular level components in plants [10]. ROS (acts as signaling molecules) is advantageous to plants under drought stress and allows them to regulate their metabolism; it also supports the acclimation process in plants during abiotic stress [98]. Nevertheless, this process can be mitigated in cells through a set of ROS-detoxifying proteins and antioxidant enzymes such as SOD, CAT, and APX [99]. Our results have shown a significant variance during the morning–afternoon loop (10 a.m. and 2 p.m.) on both day 5 and day 10 (Figure 6A). TPC has significantly elevated in spinach genotypes (Figure 5B) under drought stress; these are compounds that are naturally present in plants and are produced in the cytoplasm and endoplasmic reticulum. This plays a beneficial role as a signal molecule, scavenges the excess ROS, and also acts as a secondary antioxidant defense system under stressed conditions [95]. TPC was observed to be high in concentrations in various safflower cultivars under water stress [100]. The accumulation of TSP in leaf samples of spinach genotypes under drought stress acts as an osmotic adjustment in plants. In the present investigation, TSP in spinach leaves was not affected significantly under drought stress (Figure 5A).

With the aid of the results obtained from the photosynthetic parameter, stress indicators, photosynthetic pigments, and antioxidant activities—which have shown a significant variance between 10 a.m. and 2 p.m. (morning-afternoon loop) on both day 5 and day 10—were selected for conducting isozymes (as a proof for scavenging enzymes), protein analysis, gene expression studies, and metabolite concentration analyses. The different isoforms of antioxidant enzymes were separated using the native gel technique using an electric field where $H_2O_2$ serves as the substrate compound. In the current study, we observed the three different isoforms of SOD isozymes, which may be Mn SOD, Cu/Zn, and SOD (Figure 6B). The three different isoforms of APX may include thylakoidal APX (tAPX), stromal APX (sAPX), and cytosol APX (cAPX). The isoforms of CAT may include iron porphyrin enzymes. The chloroplast is known to be the site of photosynthesis in the leaves of green plants, and chlorophyll is utilized by chloroplast for absorbing, transferring, and transforming light energy [4]. To evaluate the effect of drought stress in spinach genotypes, a molecular-level analysis was conducted using an essential tool, chloroplast

proteomics. There was a study reported previously in Okra [48] with changes in thylakoidal proteome under drought stress, but this was neither on the basis of circadian hours nor after a re-irrigated regime. We observed a downregulated expression of thylakoidal proteins with drought-stress samples (Figure 3B) and have shown an upregulation in the expression of these photosynthetic proteins with respect to photo-periodic hours. These results have supported the results of the biochemical processes performed for the photosynthetic parameters and photosynthetic pigments.

The relative expression of the core genes of a circadian oscillator, such as *CCA1*, *LHY*, *TOC1*, and *RVE8* in spinach genotypes (MJ and DG) between 10 a.m. and 2 p.m. on day 5 (drought samples) and day 10 (re-irrigated samples) has been elucidated; and it is presented in Figure 7. The morning-phased circadian clock is expressed highly in the mesophyll tissues, whereas the evening-phased circadian clock is increasingly expressed mainly in vascular systems [101]. The two single clock-regulated Myb-like transcription factors (TFs), such as *circadian clock associated 1* (*CCA1*) and *late elongated hypocotyl* (*LHY*), are the core members of the circadian clock, which becomes expressed in the dawn period of the day [101–103]. *CCA1* and *LHY* are the morning-expressed TFs but are inhibited during the afternoon/evening by the expression of *PRR*s, mainly *TOC1* [104]; *CCA1* and *TOC1* often form an auto-regulatory positive or negative feedback loop [105]. *TOC1* (one of the small subfamilies of PRR proteins) is an evening-expressed gene that becomes activated by CCA1 or LHY [106]. *CCA1*, *LHY*, and PRRs (1, 3, 5, 7, 9) are expressed during the morning-to-evening loop, and also, they consist of core negative feedback loops [107,108]. The continuously expressed (morning to midnight) *PRR*s (1, 3, 5, 7, 9) inhibit the transcript accumulation of *CCA1*, *LHY*, and *RVE8* [101]. From a recent report, it was known that *CCA1*, *LHY*, and *RVE8* occur during dawn and reach the highest increase by the morning time period [25]. The *Reveille* (*RVE*) member *RVE8* is a homolog of *CCA1* and *LHY*, which is an essential TF that activates the evening genes by binding to the promoter regions of the evening element (EE) motif [109,110]; also, it is expressed during the afternoon time and has the target of expression similar to *CCA1* and *LHY* [110,111]. *RVE8* (Myb-like TF) is a homolog of *CCA1* and *LHY*, which becomes activated and expressed during the mid-day or afternoon hours of the circadian system [24,109].

The PRRs are a family of genes that are known to be an integral part of the circadian clock; they have a vital role in combination with the input pathway, central oscillator, and output pathway [112]. As the role of PRRs is not fully understood yet in green plants, this current study has a major focus on PRRs in spinach genotypes (MJ and DG). The relative expressions of *PRR3*, *PRR5*, *PRR7*, and *PRR9* have been observed in spinach genotypes between 10 a.m. and 2 p.m. on day 5 and day 10 (under drought stress and after re-irrigation), and it is presented on Figure 8. The PRR genes have the ability to alter the phenotypes of output processes in the circadian clock, such as hypocotyl elongation, drought-stress responses, and flowering time, which indicates their importance towards biological processes [113]. These PRRs act as transcriptional repressors towards morning-phased circadian clock genes [114,115]; also, PRRs are known as a regulatory (gating) mechanism that alters plant responses and regulates plant growth [112,116]. The *PRR3* gene provides stability for the TOC1 protein by arresting the interactions between F-box protein ZEITLUPE (ZTL) and *TOC1*; thus, the expressions of *TOC1* and *PRR3* may remain in a similar pattern. *PRR3* is a gene that is often expressed in the vasculature of cotyledons and leaves; also, it is co-regulated with *TOC1* [117]. *RVE8* binds to the promotor regions (consists of Evening Element motifs) of both *TOC1* and *PRR5*; *RVE8* and *PRR5* form a negative feedback loop [109]. *CCA1/LHY* inhibits the expressions of *PRR3*, *TOC1*, and *PRR5*, whereas it indirectly activates the transcription of *PRR7* and *PRR9* (102). The *CCA1* and *LHY* TFs directly repress the evening genes and play a crucial role in activating *PRR7*

and *PRR9* [118,119], whereas *CCA1/LHY* are repressed by the *PRR* genes directly from afternoon until midnight [114,119]. The small subfamily of proteins, such as *pseudo-response regulators* (*PRR7* and *PRR9*), are morning-phased genes, whereas *PRR5*, *PRR3*, and *PRR1* (or *TOC1* (*Timing of CAB expression 1*)) are evening-phased genes [120,121]. In spinach genotypes (MJ and DG), the core and *PRR* genes have been shown as either upregulated or downregulated expression patterns (even during their respective photo-periodic hour) on day 5 due to the effect of drought stress on plants (Figures 7 and 8). However, by day 10, after re-watering, these genes depict their expression according to their photo-periodic hours with the help of the circadian clock.

We observed upregulation in the expression levels of *DREB1*, *DREB2*, and *PIP1* under drought stress, whereas it has shown downregulation in expression patterns after re-irrigation with respect to the circadian hours (Figure 9). The *DREB* genes have shown an increased expression under drought stress in previous studies performed in Grape (*Vitis vinifera* L.) [122] and bread wheat cultivars [123]. Dehydrative responsive element binding (DREB) acts as a crucial TF [124] from the ethylene responsive factor (ERF) family (126) that can manage the expression of various stress-inducible genes [123]. There are two sub-classes of DREB, such as *DREB1/CBF* and *DREB2*; *DREB1* and *DREB2* can remarkably increase plant tolerance towards abiotic stresses, mainly water stress [125,126]. Plasma membrane intrinsic proteins (PIPs) are a subfamily of aquaporins (AQPs-drought inducible protein) [127] and drought-responsive genes, which have a main role in water permeability [128]; moreover, it has been previously reported that, in Arabidopsis, isoforms of *PIP1* have low or zero water channel activity, comparatively [128]. The upregulated expressions of *PIP* genes were observed in *Phaseolus vulgaris* plants due to the direct effect of low water content and reduction in soil water potential. It was also proposed that aquaporins have the capability to serve as osmo-sensors in plant membranes [129]. The reduction after re-watering, along with the photo-periodic hours, may indicate the role of circadian biology in controlling the effect of drought stress in spinach genotypes on day 10. Apart from the gene-regulatory network, there are several other factors that can play a vital role in maintaining defense mechanisms in plants. One of the important factors is metabolomes, in particular, with stress phytohormones that play a crucial role in maintaining drought stress, combining with gene interplays.

An intention of the analysis of targeted metabolomics by UHPLC-MS/MS was to estimate the amount of each defense hormone present in leaf samples of spinach genotypes with respect to circadian biology. With the help of this result, we were able to elucidate the concentrations of each metabolite present in each sample during the morning–afternoon loop. The defense hormones, such as JA, ABA, and SA, function as chemical signals or messengers towards environmental stresses; the stress signals provoke the initiation of a set of plant developmental and physiological processes, including stomatal closure, osmolyte accumulation, and stimulation of root growth. This further leads to a reduction in water loss and acclimation to the stressed conditions [130]. We observed an increase in the concentration of ABA in spinach genotypes under drought stress during both photo-periodic hours on day 5 (Figure 10). Under drought-stressed conditions, ABA is highly produced and accumulated in guard cells that are present around stomata; and it promotes stomatal closure, which prevents water loss and leads to water conservation [131]. The reduction of ABA after re-irrigation on day 10 on the basis of circadian biology has shown the recovery of spinach genotypes from drought stress. Plant hormones are an essential factor that helps plants withstand unfavorable conditions like drought stress [132]. SA is an important cellular regulator in plants that serves as a signaling molecule for activating the defense mechanism [133]. It has been reported in *Arabidopsis thaliana* that the accumulation of JA in leaf samples was essential for a significant rise in JA-Ile. The increase in JA and

JA-Ile is associated with an increase in cis-OPDA [134]. This indicates that JA, JA-Ile, and cis-OPDA are directly related to each other and may remain in the same pattern while being quantified, which is similar to our results that have been presented in Figure 10. These defense hormones have the ability to increase the antioxidant enzyme systems (enzymatic and non-enzymatic antioxidants), decrease membrane damage, and improve plant growth and biomass production. The quantification of ABA, JA, SA, JA-Ile, and OPDA has been reported in *Arabidopsis thaliana* and *Citrus sinensis* by LC-MS/MS in a previous study [135]. Thus, the evaluation of these metabolite concentrations in leaves of spinach genotypes (MJ and DG) during both water regimes (drought stress and re-irrigated condition) revealed that the overall recovery of phytohormones present in spinach genotypes from drought stress and mainly after re-irrigation is on the basis of photo-periodic hours. The overall schematic overview of ROS mitigation caused by drought stress is described in Figure 11.

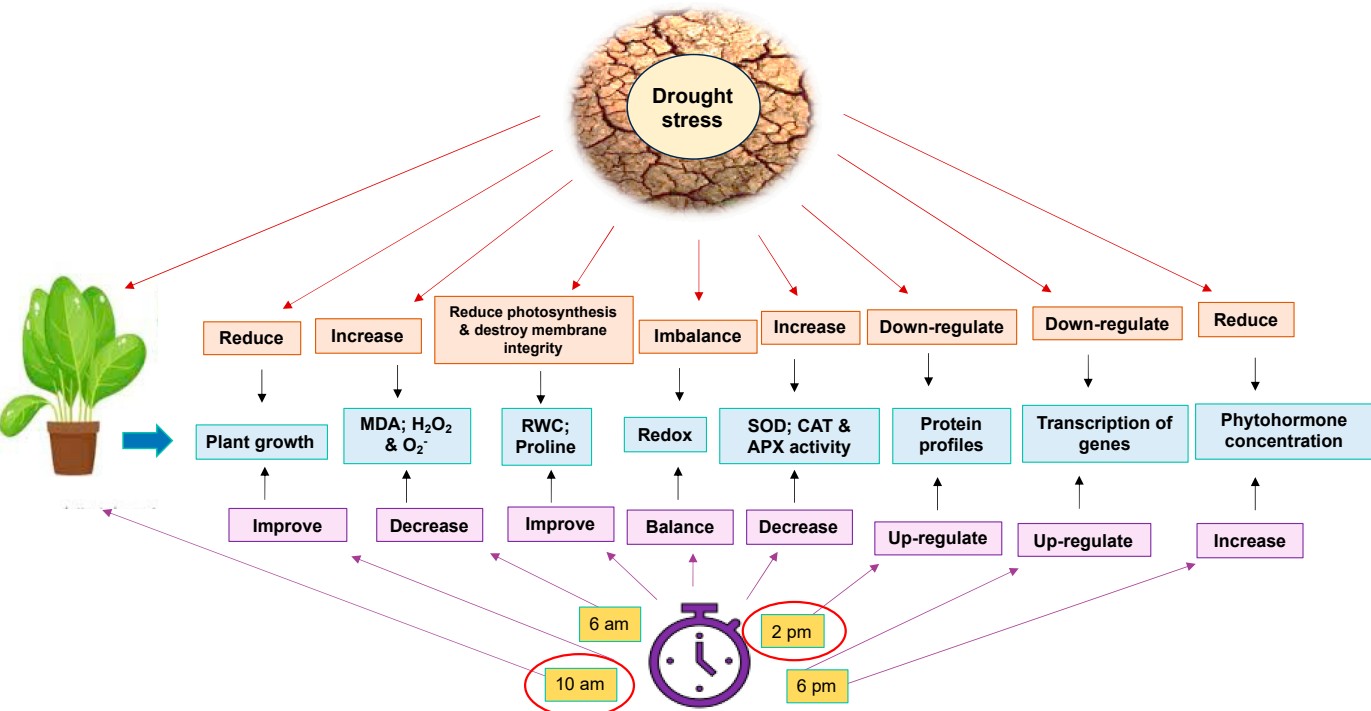

**Figure 11.** Schematic representation of drought-stress mitigation by different time periods of the circadian clock. The upward arrows show an increase, and the downward arrows show decreases in several parameters of physiology, proteins, genes, and targeted metabolome. Encircled in red indicates the time period that has a positive impact on mitigating drought stress in spinach.

## 5. Conclusions

Humans and plants alike are concerned about climate change on a global scale. It is critical to comprehend how plants react to abiotic stress, and it is imperative that this problem be resolved. Here, we have uncovered a critical function that the circadian rhythm plays in controlling how the key horticultural crop "spinach" responds to drought. Through chloroplastic controls, circadian-based morning–evening loop regulation, and reciprocal regulation between drought-responsive genes and physiological modulations, we showed how spinach modifies drought stress at a given time period. Additionally, it was determined that particular phytohormones connected to stress correspond with circadian regulatory genes in response to drought stress. Our findings also demonstrated how drought-responsive genes instruct the circadian clock genes to modify stress levels at particular time intervals. This current research provides important insights for the development of genetic strategies to improve the drought-stress tolerance of various horticulture crops.

**Author Contributions:** S.M. conceptualized and supervised the work; A.V. performed the experiments and wrote the manuscript. S.M. edited and finalized the manuscript. All authors have read and agreed to the published version of the manuscript.

**Funding:** This research was funded by VIT SEED GRANT (SG20230008).

**Institutional Review Board Statement:** Not applicable.

**Data Availability Statement:** The datasets used and/or analyzed during the current study are available from the corresponding author upon reasonable request.

**Conflicts of Interest:** The authors declare no conflicts of interest.

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
