# Peer review of "Circadian-Mediated Regulation of Growth, Chloroplast Proteome, Targeted Metabolomics and Gene Regulatory Network in Spinacia oleracea Under Drought Stress"

_agriculture, doi:10.3390/agriculture15050522_

Round 1

Reviewer 1 Report

Comments and Suggestions for Authors

The article investigates the role of the circadian clock in drought stress adaptation in spinach (Spinacia oleracea), focusing on transcriptomic, proteomic, and metabolomic responses across two genotypes. It emphasizes the impact of circadian rhythms on drought resistance, particularly through ROS scavenging, alterations in the chloroplast proteome, gene regulatory networks, and targeted metabolomics. The study is well-structured, presenting a thorough experimental approach and offering valuable insights into circadian-mediated drought stress adaptation. However, the manuscript would benefit from refinements in scientific clarity, data interpretation, and figure accuracy to enhance its impact. Below are my comments for improving the manuscript to make it suitable for publication.

1.     In abstract the phrase "scrutinize the beneficial role of the circadian clock in attributing drought stress" is ambiguous revise to accurately convey meaning.

2.     Key quantitative data is missing in the abstract; including numerical results would strengthen impact.

3.     The introduction is lengthy and should be more focused on the research gap rather than general drought stress effects.

4.     The sample size and experimental replicates need explicit clarification to enhance reproducibility.

5.     The method for chlorophyll fluorescence measurement lacks details on measurement duration and standardization.

6.     Explain why only 10 AM and 2 PM samples were selected for molecular analyses justify in relation to circadian rhythm phases.

7.     The RNA isolation and RT-PCR methodology need information on RNA quality control measures.

8.     Provide more information on how targeted metabolomics was normalized between samples.

9.     The results use inconsistent terminology (e.g., "significant increase" vs. "peak") standardize the language.

10.  Statistical significance indicators are missing from some figures add significance bars where applicable.

11.  The stomatal structure micrographs require scale bars for accurate interpretation.

12.  Data presentation of oxidative stress markers (MDA, Hâ‚‚Oâ‚‚, O₂ˉ) should be aligned in a comparative manner.

13.  The BN-PAGE protein bands need molecular weight markers for reference.

14.  The text describing antioxidant enzyme activities contradicts certain figure interpretations ensure consistency.

15.  The qRT-PCR expression data lacks validation with additional replicates mention if biological triplicates were averaged.

16.  The circadian clock’s role in ROS scavenging is not mechanistically explained clarify with a model.

17.  Certain speculative statements (e.g., "circadian clock entirely controls stress tolerance") should be tempered with supporting evidence.

18.  The interplay between metabolomics and gene regulation needs deeper integration in the discussion.

19.  Figure 1 lacks high-resolution images; improve clarity.

20.  Figure 2B does not clearly differentiate between control and treatment groups use distinct markers.

21.  Tables presenting qRT-PCR primers need information on amplicon sizes.

22.  The stomatal index graph should include representative microscopic images of stomata from both genotypes.

23.  Some graphs use inconsistent color schemes harmonize for readability.

Comments on the Quality of English Language

The manuscript's English requires improvement to enhance clarity, readability, and scientific precision. Several sentences are overly complex or ambiguous, making key findings difficult to interpret. Grammatical inconsistencies and awkward phrasing appear throughout the text, particularly in the abstract and discussion sections. Additionally, technical terms should be used more precisely to ensure accuracy. A thorough language revision, preferably by a native English-speaking editor or professional proofreading service, is recommended to improve fluency and coherence.

Author Response

Reviewer 1

The article investigates the role of the circadian clock in drought stress adaptation in spinach (Spinacia oleracea), focusing on transcriptomic, proteomic, and metabolomic responses across two genotypes. It emphasizes the impact of circadian rhythms on drought resistance, particularly through ROS scavenging, alterations in the chloroplast proteome, gene regulatory networks, and targeted metabolomics. The study is well-structured, presenting a thorough experimental approach and offering valuable insights into circadian-mediated drought stress adaptation. However, the manuscript would benefit from refinements in scientific clarity, data interpretation, and figure accuracy to enhance its impact. Below are my comments for improving the manuscript to make it suitable for publication.

  1. In abstract the phrase "scrutinize the beneficial role of the circadian clock in attributing drought stress"is ambiguous revise to accurately convey meaning.

We have revised the phrase in the revised manuscript as suggested by reviewer

  1. Key quantitative data is missing in the abstract; including numerical results would strengthen impact.

Key quantitative data has been given in the revised manuscript, please see highlighted areas in the abstract  

  1. The introduction is lengthy and should be more focused on the research gap rather than general drought stress effects.

We respect reviewer suggestion to shorten the introduction, but sections for drought stress must be also given to understand about it followed by research gap that is given in second section of introduction section. Hence, we have not shorten the introduction section

  1. The sample size and experimental replicates need explicit clarification to enhance reproducibility.

The sample size and experimental replicate details are given in methodology section, please see highlighted sections

  1. The method for chlorophyll fluorescence measurement lacks details on measurement duration and standardization.

The method of PSII quantum yield is given in methodology section as suggested by reviewer. Please see the highlighted section

  1. Explain why only 10 AM and 2 PM samples were selected for molecular analyses justify in relation to circadian rhythm phases.

We actually have analysed in previous studies that 10AM and 2PM is best to alleviate salinity stress (https://www.mdpi.com/2077-0472/13/2/429). Therefore we have selected these 2 time points that is already discussed in discussion section too

  1. The RNA isolation and RT-PCR methodology need information on RNA quality control measures.

The RNA isolation quality information has been provided in the methodology section of RNA isolation (highlighted) that were used to synthesize cDNA and for RT-PCR further

  1. Provide more information on how targeted metabolomics was normalized between samples.

The normalization of samples were done by using the known standard samples, the information is provided in the revised section of targeted metabolomics

  1. The results use inconsistent terminology (e.g., "significant increase""peak") standardize the language.

We have standardized the terminologies wherever, required as suggested by reviewer, and all changes are  highlighted  

  1. Statistical significance indicators are missing from some figures add significance bars where applicable.

We have corrected them, due to change in format of manuscript it had initially been missed out at some places

  1. The stomatal structure micrographs require scale bars for accurate interpretation.

We have added scale bars in the revised manuscript

  1. Data presentation of oxidative stress markers (MDA, H₂O₂, O₂ˉ) should be aligned in a comparative manner.

We have tried to align them in the revised manuscript

  1. The BN-PAGE protein bands need molecular weight markers for reference.

BN-PAGE is a native form of gel electrophoresis, and protein marked cannot be run, with samples.

Usually, reference papers are used for identifying the type/name of protein

  1. The text describing antioxidant enzyme activities contradicts certain figure interpretations ensure consistency.

We have tried to make consistency in certain parts wherever, necessary

  1. The qRT-PCR expression data lacks validation with additional replicates mention if biological triplicates were averaged.

Yes, biological triplicates were taken and it has been mentioned in methodology section as well as results sections (highlighted)

  1. The circadian clock’s role in ROS scavenging is not mechanistically explained clarify with a model.

We have given a machinal model to explain the ROS scavenging activity at end of discussion before conclusion section Figure 11

  1. Certain speculative statements (e.g., "circadian clock entirely controls stress tolerance") should be tempered with supporting evidence.

We have not made entirely a speculation that circadian clock controls stress tolerance, but we have made speculation that circadian clock can help in stress tolerance particularly in discussion sections where proper references are also provided

  1. The interplay between metabolomics and gene regulation needs deeper integration in the discussion.

We have tried to make corelation of genes and metabolomics as suggested by reviewer highlighted

  1. Figure 1 lacks high-resolution images; improve clarity.

Provided, with high clarity

  1. Figure 2B does not clearly differentiate between control and treatment groups use distinct markers.

     High resolution provided, to distinctly differentiate between control and stress

  1. Tables presenting qRT-PCR primers need information on amplicon sizes.

Amplicon size is given in the revised manuscript

  1. The stomatal index graph should include representative microscopic images of stomata from both genotypes.

Microscopic and stomatal index graphs is represented from both genotypes

  1. Some graphs use inconsistent color schemes harmonize for readability.

We have tried to increase the resolution of all figures to make them properly readable

Editor

The manuscript's English requires improvement to enhance clarity, readability, and scientific precision. Several sentences are overly complex or ambiguous, making key findings difficult to interpret. Grammatical inconsistencies and awkward phrasing appear throughout the text, particularly in the abstract and discussion sections. Additionally, technical terms should be used more precisely to ensure accuracy. A thorough language revision, preferably by a native English-speaking editor or professional proofreading service, is recommended to improve fluency and coherence.

Authors have requested one of the collaborator native English speaker to help in preferable changes in grammatical errors, besides, we have used Grammarly tool to edit

Reviewer 2 Report

Comments and Suggestions for Authors

The manuscript entitled Circadian-mediated regulation of growth, chloroplast proteome, targeted-metabolomics and gene-regulatory network in Spinacia oleracea under drought stress provides information regarding the role of circadian clock in attributing drought stress in spinach. This study seems interesting; however, I have a few comments to share.

The authors did not mention why they selected the genotypes “Malav Jyoti” and “Delhi Green. They have to describe the drought-related characteristics of the genotypes.

Why authors did not use soil water content (control, moderate, severe) to check the drought stress effects.

What was the reason to select some responsive genes to study their expression? Why the authors did not evaluate transcriptomic analysis to obtain comprehensive findings.

How many times was the experiment repeated? added to the text.

The quality of figures is not acceptable.

The discussion is too long and mostly narrative. The authors need to compare the findings of their study with those presented in the published papers.

How the results of this study could develop genetic strategies to improve the drought stress tolerance.

The authors conducted the effect of circadian clock on growth, chloroplast proteome, targeted-metabolomics and gene-regulatory network in Spinacia oleracea under drought stress and got a wide number of results. However, the question is how we might apply these findings within greenhouse or agricultural environments.

Author Response

Reviewer 2

The manuscript entitled Circadian-mediated regulation of growth, chloroplast proteome, targeted-metabolomics and gene-regulatory network in Spinacia oleracea under drought stress provides information regarding the role of circadian clock in attributing drought stress in spinach. This study seems interesting; however, I have a few comments to share.

The authors did not mention why they selected the genotypes “Malav Jyoti” and “Delhi Green. They have to describe the drought-related characteristics of the genotypes.

Authors have mentioned why Malav Jyoti and Delhi Green was selected for the study in the revised manuscript. Please see the highlighted methodology section

Why authors did not use soil water content (control, moderate, severe) to check the drought stress effects.

Authors have analysed the effects of drought stress by weighing the bags and moisture sensors, the information is given in the revised manuscript please see the highlighted section of methodology  

What was the reason to select some responsive genes to study their expression? Why the authors did not evaluate transcriptomic analysis to obtain comprehensive findings.

Since, the specific genes for circadian clock are known for their functions like flowering, or any other major pathways thus selective genes were analysed and similarly genes known for maintaining drought and water relations genes were selected.

For your kind information, we are currently working on transcriptomic part, that will be represented as a sperate data due to huge number of  data

How many times was the experiment repeated? added to the text.

Mostly, all the experiments were carried out 3 times, (repeated)

The quality of figures is not acceptable.

Due to tiff formatting the quality of original figures are getting very less dpi. We have tried to upload a high resolution figures in revised manuscript  

The discussion is too long and mostly narrative. The authors need to compare the findings of their study with those presented in the published papers.

Well, we have tried to make sure that related studies given are referred and also for each experiment separately that can give us better idea about all the works carried out

How the results of this study could develop genetic strategies to improve the drought stress tolerance.

Because genetic strategy includes, gene therapy, gene modulations, and other sequencing methods.

In our study we have used gene modulations, thus our study can be used for further used how to control gene expression at particular time points, or a base for transcriptomics study like you have suggested in your above comments

The authors conducted the effect of circadian clock on growth, chloroplast proteome, targeted-metabolomics and gene-regulatory network in Spinacia oleracea under drought stress and got a wide number of results. However, the question is how we might apply these findings within greenhouse or agricultural environments.

Firstly the study was conducted in protected structures, allowing plants to optimize their physiological processes throughout the day, maximizing photosynthesis, growth rate, and overall productivity by aligning their activities with the natural light-dark cycle, essentially "anticipating" environmental changes like light and temperature fluctuations, leading to better adaptation and improved yield when properly managed. Thus, even if the plants are under any abiotic stress the current study has depicted that it can be managed with the help of clock

Reviewer 3 Report

Comments and Suggestions for Authors

The manuscript is well-written. I’ve mentioned a few suggestions. Please incorporate them and resubmit it.

Line 33: Please provide a range of agriculture production which was specifically affected by drought.

Line 54: Could you create a figure indicating all the mentioned plant processes and how drought changes them?

Line 273: Could you please clarify what the author means when they say “replicate means were compared using students t-test”? Additionally, could you explain the difference between a student t-test and a Tukey test, and when one would be preferred over the other? Finally, please provide the model statement used in SAS.

Author Response

Reviewer 3

The manuscript is well-written. I’ve mentioned a few suggestions. Please incorporate them and resubmit it.

We thank reviewer for her/his comments and suggestion and we have incorporated them

 Line 33: Please provide a range of agriculture production which was specifically affected by drought.

 Range for crop reduction by drought is given in the revised manuscript

Line 54: Could you create a figure indicating all the mentioned plant processes and how drought changes them?

Yes, figure has been drawn as suggested, please see figure 11 in the revised manuscript

Line 273: Could you please clarify what the author means when they say “replicate means were compared using students t-test”? Additionally, could you explain the difference between a student t-test and a Tukey test, and when one would be preferred over the other? Finally, please provide the model statement used in SAS.

Replicate means, all three biological samples that were taken for experimental analysis and data was collected and were used for statistical analysis for mean and SE.

A Student t-test is used to compare the means of two groups, while a Tukey test (also known as Tukey's HSD) is used to compare the means of multiple groups to each other after a significant ANOVA result, essentially performing multiple pairwise comparisons with a correction for the increased chance of Type I error when making many comparisons simultaneously; meaning, a Tukey test is used for post-hoc analysis following ANOVA, while a t-test is a standalone test for comparing just two groups. Since we had many treatments and comparisons we had to use both t-test and tukeys t test for  ANNOVA analysis

Model statement is also provided in the revised manuscript.

Round 2

Reviewer 1 Report

Comments and Suggestions for Authors

The revised manuscript has significantly improved, and you have successfully addressed all my concerns. The clarity, scientific rigor, and overall quality of the work have been greatly enhanced. I appreciate your efforts in refining the manuscript, which now meets the required standards. I find it suitable for publication in its current form.